# OCEANFILMS sea spray organic aerosol emissions – Implementation in a global climate model and impacts on clouds

Susannah M. Burrows[1], Richard Easter[1], Xiaohong Liu[2], Po-Lun Ma[1], Hailong Wang[1], Scott M. Elliott[3], Balwinder Singh[1], Kai Zhang[1], and Philip J. Rasch[1]

[1]Pacific Northwest National Laboratory, Richland, Washington, USA
[2]Texas A&M University, College Station, Texas, USA
[3]Los Alamos National Laboratory, Los Alamos, New Mexico, USA

*Correspondence to:* Susannah M. Burrows (susannah.burrows@pnnl.gov)

**Abstract.** Sea spray aerosol is one of the major sources of atmospheric particulate matter globally. It has increasingly been recognized that organic matter derived from ocean biological precursors contributes significantly to the composition of submicron sea spray and may modify sea spray aerosol impacts on clouds and climate. This paper describes the implementation of the OCEANFILMS parameterization for sea spray organic aerosol emissions in a global Earth system model, the Energy Exascale Earth System Model (E3SM). OCEANFILMS is a physically based model that links sea spray chemistry with ocean biogeochemistry using a Langmuir partitioning approach. We describe the implementation details of OCEANFILMS within E3SM, compare simulated aerosol fields with observations, and investigate impacts on simulated clouds and climate. Four sensitivity cases are tested, in which organic emissions either strictly add to or strictly replace sea salt emissions (in mass and number), and are either fully internally or fully externally mixed with sea salt. The simulation with internally mixed, added organics agrees reasonably with observed seasonal cycles of organic matter in marine aerosol, and has been selected as the default configuration of E3SM. In this configuration, sea spray organic aerosol contributes an additional source of cloud condensation nuclei, adding up to 30 cm$^{-3}$ to Southern Ocean boundary-layer cloud condensation nuclei concentrations (supersaturation= 0.1%). The addition of this new aerosol source strengthens shortwave radiative cooling by clouds by $-0.36$ W/m$^2$ in the global annual mean, and contributes more than $-3.5$ W/m$^2$ to summertime zonal mean cloud forcing in the Southern Ocean, with maximum zonal mean impacts of about $-4$ W/m$^2$ around 50°S–60°S. This is consistent with a previous top-down, satellite-based empirical estimate of the radiative forcing by sea spray organic aerosol over the Southern Ocean. Through its mechanistic approach, OCEANFILMS offers a path towards improved understanding of the feedbacks between ocean biology, sea spray organic matter, and climate.

## 1 Introduction and motivation

It has long been noted that organic matter constitutes a substantial portion of submicron marine aerosol mass (Hoffman and Duce, 1974, 1976, 1977; Duce et al., 1983; Oppo et al., 1999). However, it has only recently been widely appreciated that water-insoluble organic matter (WIOM) contributes substantially to submicron marine aerosol downwind of strong seasonal

phytoplankton blooms (O'Dowd et al., 2004), and has the potential to affect the number and chemical character of aerosol and cloud condensation nuclei (CCN) in certain marine regions (McCoy et al., 2015).

Although organic matter in ambient sea spray can also arise from condensation of volatile organic compounds (i.e., secondary organic aerosol), a large body of experimental evidence shows that nascent sea spray also contains organic matter of biogenic origin. Experiments using physical simulations of the sea spray aerosol production process have shown that organic matter is co-emitted into sea spray aerosol together with salt (Keene et al., 2007; Facchini et al., 2008a; Gao et al., 2012; Bates et al., 2012; Schmitt-Kopplin et al., 2012; Ault et al., 2013; Prather et al., 2013; Quinn et al., 2014; Frossard et al., 2014a; Long et al., 2014; Alpert et al., 2015; Kieber et al., 2016). The primary sea spray origin of marine organic matter is also supported by laboratory studies and field experiments using a variety of analytical methods (see review by Frossard et al., 2014b), which show that the chemical composition of submicron marine aerosol is similar to material drawn from the sea surface microlayer (Facchini et al., 2008b; Leck and Bigg, 2005b; Russell et al., 2010).

Clouds in remote marine areas are particularly sensitive to changes in aerosol concentrations (Karydis et al., 2012; Moore et al., 2013), because cloud droplet number concentrations (CDNCs) respond to perturbations in aerosol more strongly when background aerosol concentrations are low (Pringle et al., 2012). These clouds, located in regions where anthropogenic aerosols are scarce, are primarily influenced by natural aerosol sources (Hamilton et al., 2014), and the ability to constrain climate sensitivity using historical climate records is in part limited by quantification of these natural sources (Karydis et al., 2012; Carslaw et al., 2013; Regayre et al., 2020). Recent research also suggests that sea spray organic aerosol (which we abbreviate as MOA, for marine organic aerosol) can serve as nuclei for freezing of cloud droplets (Knopf et al., 2011; Wilson et al., 2015; DeMott et al., 2016) and may play an important role as atmospheric ice nuclei in remote marine regions (Schnell and Vali, 1975, 1976; Burrows et al., 2013a; Wilson et al., 2015; Vergara-Temprado et al., 2017; McCluskey et al., 2019; Zhao et al., 2021).

## 1.1 Existing model parameterizations of MOA emissions

Several previous studies have proposed and implemented representations of MOA emissions within a global climate model, with variations in the emissions representation and the assumed aerosol microphysical properties. Most of these studies were based on aerosol chemistry observations taken primarily from two sites, at Mace Head, Ireland and at Amsterdam Island, in the Southern Ocean (Sciare et al., 2009a; Rinaldi et al., 2013). O'Dowd et al. (2008) proposed an early parameterization for MOA emissions, which was later modified by Langmann et al. (2008) for inclusion in a global model, and further updated and evaluated by Vignati et al. (2010) (which also corrects a typographical error in the formulation as printed in Langmann et al., 2008). In this approach, the organic mass fraction (OMF), defined as the ratio of MOA mass to the total of MOA and sea salt aerosol (SSA) mass, depends linearly on chlorophyll-a [Chl-a], with an imposed upper bound corresponding to the highest observed values of OMF. Rinaldi et al. (2013) further updated this parameterization by adding a linear dependence of OMF on wind speed. This study also showed that the correlation between upwind [Chl-a] and OMF was improved when a time lag of 8–10 days was introduced.

Gantt et al. (2011) proposed an emission parameterization in which OMF depends on wind speed and [Chl-*a*], by fitting a nonlinear equation form to observed OMF at Mace Head, Ireland and Point Reyes, California. Meskhidze et al. (2011) further evaluated this parameterization and compared it to Vignati et al. (2010), concluding that both parameterizations captured the magnitude of MOA concentrations, with Gantt et al. (2011) attaining better seasonality.

Long et al. (2011) proposed another alternative approach in which OMF depends nonlinearly on [Chl-*a*], using the functional form of the Langmuir isotherm to drive the relationship, and using a fit to observed [Chl-*a*] and OMF from two sea spray generation experiments to constrain the model parameters. The parameterization also takes particle diameter into consideration.

Additional model studies of MOA have largely built upon these initial proposed parameterizations and further explored their uncertainties, their sensitivities to certain aspects of the model implementation, and the resulting implications for climate. These studies have emphasized uncertainties in the choice to either add to or replace existing sea salt (Westervelt et al., 2012), to assume that aerosols are internally versus externally mixed (Meskhidze et al., 2011), and in the sea spray fine mode particle size (Tsigaridis et al., 2013).

## 1.2 Estimates of global annual MOA emissions

The best estimates for global emissions of primary marine organic matter (OM) from different model studies span a range from at least $2.8$–$76$ Tg y$^{-1}$ (fine mode emissions) and $4.5$–$34.9$ Tg y$^{-1}$ (coarse mode emissions) (Spracklen et al., 2008a; Gantt et al., 2009; Vignati et al., 2010; Myriokefalitakis et al., 2010; Ito and Kawamiya, 2010; Long et al., 2011; Gantt et al., 2011; Myriokefalitakis et al., 2010; Westervelt et al., 2012; Tsigaridis et al., 2013; Vergara-Temprado et al., 2016). These studies used both different emissions parameterizations and different host model systems (e.g., with differences in aerosol parameterizations, and atmospheric physics parameterizations impacting transport and removal), which are likely to also cause differences in the simulated atmospheric residence time of MOA in each model. We have used an organic matter to organic carbon mass ratio (OM:OC) of 1.8 to convert where OC values have been reported; 1.8 is the OM:OC ratio of Suwanee River fulvic acid, which has been used as a proxy for marine organic matter. In the few measurements of OM:OC ratios that have been conducted for marine boundary layer aerosol organic matter, observed values range from at least 1.2 to 2.1 (Russell, 2003; Hawkins et al., 2010; Saliba et al., 2020). The calculation of underlying spray emission fluxes also contributes a significant source of uncertainty, with parameterizations differing by as much as a factor of two (De Leeuw et al., 2011), which further amplifies uncertainties in MOA emissions (Tsigaridis et al., 2013).

Other studies have aimed to constrain the magnitude of the global MOA source required to produce the best agreement with observed concentrations. Lapina et al. (2011) found that adding a MOA source of about 9 TgC y$^{-1}$ to simulations in a global atmospheric chemistry model improved agreement with remote ship-based observations of MOA concentrations from multiple field campaigns. Spracklen et al. (2008b) used observed oceanic organic carbon (OC), back-trajectories, and remotely sensed [Chl-*a*] from Mace Head, Amsterdam Island and the Azores to derive an empirical relationship between [Chl-*a*] and total (primary and secondary) oceanic organic aerosol concentrations. This study found that including an oceanic OC source of ca. 8 Tg/yr improved the modeled seasonal cycle at Mace Head and Amsterdam Island, and increased the global burden of OC by 20%, and by up to a factor of 20 or more in parts of the Southern Ocean.

## 1.3 Need for a mechanistic parameterization and aims of this study

While empirical, [Chl-*a*]-based parameterizations have been successful in capturing some major observed features of the organic fraction of sea spray aerosol and its seasonal cycle, particularly at locations like Mace Head, Ireland, and Amsterdam Island, these approaches do not offer a path towards explaining or testing hypotheses to explain the seasonal and geographic variability in the emissions of organic matter in sea spray. In particular, without understanding the mechanisms driving these emissions, we cannot have confidence that empirical parameterizations derived from mid-latitude observations will be an accurate guide to the behavior of tropical or polar ocean ecosystems, or that present-day observations will be an accurate guide to the behavior of future ocean ecosystems. In an effort to provide a path forward, Elliott et al. (2014) proposed the prospect of an approach based on understanding of ocean surface films, and Burrows et al. (2014) introduced a new framework for modeling the functional relationships between ocean biogeochemical variables and the composition of emitted sea spray particles, called OCEANFILMS (**O**rganic **C**ompounds from **E**cosystems to **A**erosols: **N**atural **F**ilms and **I**nterfaces via **L**angmuir **M**olecular **S**urfactants). OCEANFILMS describes the organic mass fraction of emitted sea spray aerosol as a function of several classes of marine organic matter, each of which is assigned several chemical characteristics: adsorptivity at the air–water interface, molecular weight, area occupied at the air–water interface, and organic matter to organic carbon mass ratio (OM:OC). The value of each of these parameters is derived from laboratory studies of selected surrogate molecules, as described in detail in Burrows et al. (2014); the ocean distributions of surfactants are described further in Ogunro et al. (2015).

To further investigate the potential impacts of MOA on aerosol concentrations and chemistry, CCN, and clouds, we implemented the OCEANFILMS parameterization into an early development version of a global Earth system model, the Energy Exascale Earth System Model (E3SM). Here we evaluate the simulated aerosol number and mass concentrations, and chemistry, with respect to in situ observations, and examine the climate implications of MOA and its sensitivity to assumptions about its mixing state with sea salt and about whether it adds to or replaces existing sea salt emissions.

## 2 Implementation of OCEANFILMS in E3SM

### 2.1 Description of the E3SM Atmosphere Model

E3SM is a global Earth system model developed by the U.S. Department of Energy (DOE) for high-resolution modeling on leadership supercomputing facilities (Golaz et al., 2019). The model is a descendant of the Community Earth System Model version 1 (CESM1; Hurrell et al., 2013).

This study uses an early, pre-release version of the E3SM Atmosphere Model (EAM), which is a descendant of the CAM5 (Community Atmosphere Model 5) model (Neale et al., 2010). The EAM version used here closely resembles CAM5.3, except for the use of the MAM4 aerosol microphysics in place of the default MAM3 microphysics, some modifications to the model's treatments of aerosol microphysics and aerosol–cloud interactions, which have been documented in previous publications and are summarized in the following paragraph, and some minor bug fixes and retuning that have only small impacts on the simulated climate.

The implementation of OCEANFILMS described here builds on the 4-mode version of MAM (MAM4, Liu et al., 2016), which is the default aerosol model in E3SMv1 (Wang et al., 2020). MAM4 is an extension of MAM3, the 3-mode Modal Aerosol Microphysics (Liu et al., 2012), which is the default aerosol microphysics in CAM5.3. MAM3 represents the aerosol size distribution by means of three lognormal modes; MAM4 extends this treatment by adding a fourth, insoluble submicron aerosol mode (the primary carbon mode), which carries primary organic carbon (POC) and black carbon (BC) aerosols. This modification significantly improves the simulated concentrations of POC and BC relative to MAM3 simulations, at a lower computational cost than a more detailed 7-mode treatment. The impact of MAM4 on simulated aerosol in CAM5.3 is described in Liu et al. (2016).

Specific refinements to the MAM aerosol treatments used herein, and evaluations of simulated aerosol species with respect to observations, have been documented in a series of prior publications. Wang et al. (2013) documents a number of improvements to representations of aerosol–cloud interactions that improve simulated remote aerosol concentrations in remote regions and the mid- to upper troposphere, particularly for BC aerosol. Simulation of sulfate aerosol is discussed in greater detail in Yang et al. (2017). Biomass burning aerosol is the focus of Das et al. (2017), which compares CAM5 with several other global models. Aerosol lifetimes are evaluated and compared with 18 other global models in Kristiansen et al. (2016).

The simulation of aerosol indirect effects in variants of CAM5 is documented and discussed in detail in several previous papers (Ghan et al., 2016; Zhang et al., 2016; Gryspeerdt et al., 2017). Ghan et al. (2016) developed a framework for calculating aerosol indirect effects as the result of a chain of contributing response functions, and quantifies the strength of each term for nine different global models, including both CAM5.3 and the Pacific Northwest National Laboratory (PNNL) configuration of CAM5.3 that includes the modifications described above (i.e., MAM4 and modifications to aerosol–cloud processes). Gryspeerdt et al. (2017) documents the sensitivity of cloud-top droplet number concentration to various aerosol proxies, such as CCN number (at $S = 0.3\%$ supersaturation) at 1 km, for CAM5.3 and other models.

For clarity, we would like to alert readers that the atmosphere model in the released version of E3SMv1 differs in several key respects from the early pre-release version of E3SM used in this study. Notably, in E3SMv1, the cloud parameterization has been replaced with the CLUBB (Cloud Layers Unified By Binormals) scheme, and the number of vertical layers has been increased from 30 layers to 72 layers. Aerosols, clouds, and aerosol–cloud interactions in E3SMv1 have been described evaluated elsewhere (Xie et al., 2018; Golaz et al., 2019; Zhang et al., 2019; Wang et al., 2020).

In summary, the aerosols and clouds in CAM5.3, the modifications that are included here in EAM, and the impacts of those modifications on the simulated aerosol and aerosol–cloud interactions, have been documented and discussed in detail in the previous studies referenced above. Consequently, we do not extensively evaluate the model's aerosols and clouds here. Instead, we focus on the addition of a source of sea spray aerosol organic matter and its impact on simulated sea spray aerosol chemistry and clouds.

## 2.2 Introduction of MOA tracers

The unmodified MAM4 model carries the following chemical species: sea salt, dust, sulfate, SOA, BC, and all non-MOA primary organic aerosol (POA), e.g., from terrestrial combustion sources and ship emissions. Each species is characterized by

**Table 1.** Aerosol species and material properties used in the model simulations

| Abbreviation | Name | Density [kg m$^{-3}$] | Hygroscopicity ($\kappa$) |
|---|---|---|---|
| MOA | MOA | 1,601 | 0.1 |
| NCL | Sea salt | 1,900 | 1.16 |
| POA | Primary organic matter | 1,000 | $1.0 \times 10^{-10}$ |
| SOA | Secondary organic matter | 1,000 | 0.14 |
| SO4 | Sulfate aerosol | 1,000 | 0.507 |
| DST | Dust | 2,600 | 0.068 |
| BC | Black carbon | 1,700 | $1.0 \times 10^{-10}$ |

**Table 2.** MAM4 modes, their size parameters and tracers carried, including number ($N$) and species mass $M_{\text{species}}$.

| | Size range | $\sigma_g$ | Nominal $D_{gn}$ [m] | Low bound $D_{gn}$ [m] | High bound $D_{gn}$ [m] | Species |
|---|---|---|---|---|---|---|
| Aitken | 20–80 nm | 1.6 | 2.6e-8 | 8.70e-9 | 5.20e-8 | $N, M_{SO_4}, M_{SOA}, M_{NCL}, M_{MOA}$ |
| Accumulation | 80 nm–1 μm | 1.8 | 1.1e-7 | 5.35e-8 | 4.40e-7 | $N, M_{SO_4}, M_{SOA}, M_{POA}, M_{BC}$, $M_{DST}, M_{NCL}, M_{MOA}$ |
| Coarse | 1–10 μm | 1.8 | 2.0e-6 | 1.00e-6 | 4.00e-6 | $N, M_{SO_4}{}^{1}, M_{SOA}{}^{1}, M_{POA}{}^{1}, M_{BC}{}^{1}$, $M_{DST}, M_{NCL}, M_{MOA}$ |
| Primary carbon | 80 nm–1 μm | 1.6 | 5.0e-8 | 1.00e-8 | 1.00e-7 | $N, M_{POA}, M_{BC}, M_{MOA}$ |

[a]The original formulation of MAM4 did not contain $M_{SO_4}$ $M_{BC}$, $M_{SOA}$, and $M_{POA}$ in the coarse mode (Liu et al., 2016). These species were added to the coarse mode as part of the resuspension treatment discussed in this paper, as mass from any species can be transferred to the coarse mode during resuspension.

physical properties describing its optical properties, density, and hygroscopicity, summarized in Table 1. POA is also referred to as primary organic matter (POM). The chemical species carried in each mode of the MAM4 are identified in Table 2. In the model version used here, the coarse mode also contains BC, SOA, and POA. Each mode's lognormal size distribution is defined by its prognostic aerosol number and mass mixing ratios and a fixed geometric standard deviation ($\sigma_g$) (Table 2), and the mode's number-median diameter ($D_{gn}$) is a diagnostic variable.

In order to fully represent primary MOA in the model and allow for the specification of chemical properties particular to this class of particles, we introduced an additional aerosol chemical species into the model, which we term "MOA." MOA tracers were introduced into the model in each of the MAM4 aerosol modes.

## 2.3 Sea spray emissions

Sea spray aerosol in MAM is emitted according to the parameterization of Mårtensson et al. (2003) for particle diameters from 20 nm–2.5 μm, and Monahan (1986) from 2.5–10 μm. The Mårtensson et al. parameterization is based on laboratory simulations of particle production, using a sintered glass filter to generate a bubble plume leading to bubble bursting and emissions. These experiments used synthetic seawater, i.e., pure Milli-Q water with the addition of synthetic sea salt. Synthetic sea salt concentrates also contain trace amounts of dissolved organic carbon Arnold et al. (2007), although their representativeness for natural seawater is unclear. The Monahan (1986) parameterization, by contrast, was derived from whitecap simulation experiments in a tank filled with natural seawater collected from open coastal waters.

## 2.4 Emissions of MOA according to OCEANFILMS

The OCEANFILMS parameterization, introduced and described in detail in Burrows et al. (2014), proposes a mechanistic approach for connecting emissions of MOA to models of ocean biogeochemistry. As described previously in Burrows et al. (2014), we used the Parallel Ocean Program (POP; Maltrud et al., 1998) to simulate the ocean's general circulation and its biogeochemical elemental cycling (BEC) routines (Moore et al., 2004) to simulate marine biogeochemistry. Both are components of the Community Earth System Model (CESM; Hurrell et al., 2013; UCAR, 2021). Calculations of the ocean biogeochemistry fields were performed using the CESM 1.0 beta release 11. Because it uses prescribed input files obtained from simulations performed with the earlier POP ocean model, rather than online-simulated biogeochemistry fields, the current implementation of OCEANFILMS in E3SM is not affected by the large biases in prediction of ocean biogeochemistry that have been documented in the first release version of E3SM (Burrows et al., 2020).

Monthly-mean concentrations of five broad classes of macromolecules in ocean surface waters are derived from the POP-simulated distributions of phytoplankton, zooplankton, and semi-labile dissolved organic carbon, and are provided to E3SM through prescribed input files. The files containing these macromolecular distributions are publicly available as part of the E3SM input data repository (see data availability statement).

Chemical and physical properties are assigned to each of these macromolecular classes, based on representative proxy molecules for which laboratory measurements are available. Using a Langmuir isotherm-based approach, OCEANFILMS then predicts the surface coverage of ocean bubble films with each of these model macromolecules. This surface film coverage, together with a prescribed bubble film thickness, determines the OMF of the emitted sea spray aerosol, which is calculated online within E3SM on the basis of the prescribed macromolecule distributions.

While OCEANFILMS predicts the OMF solely as a function of the prescribed macromolecule fields, the amount of emitted MOA depends on the combination of the OMF with the emitted sea spray, which is a function of wind speed and sea surface temperature. The application of OMF to the sea spray emissions requires additional assumptions regarding the mixing state and the impact of organic emissions on total emitted particle number and mass, which we explore in four sensitivity cases (described in Section 3.1).

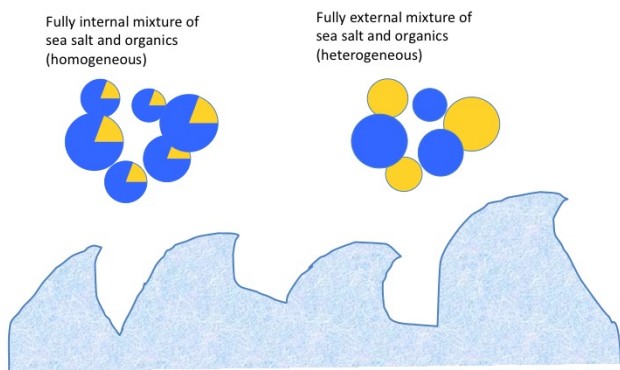

**Figure 1.** Illustration of internal versus external mixing states of sea spray aerosol upon emission. The emitted aerosol contains both sea salt (blue) and organic matter (yellow).

## 2.5 Participation of MOA in transport, aerosol and cloud microphysical processes, and loss processes

Aerosol particles evolve through a large number of processes including transport (by resolved winds, turbulent mixing, convective cloud updrafts and downdrafts, gravitational sedimentation, or dry deposition), emissions, microphysical processes (condensation and evaporation of trace gases [including water vapor], homogeneous nucleation, coagulation, aging), and cloud or precipitation processes (aqueous chemistry in cloud droplets, activation, resuspension from evaporating cloud droplets and rain, in-cloud and below-cloud wet removal by both stratiform and convective clouds, or precipitation). MOA participates in almost all of these processes within the model.

MAM assumes that within each mode, particles are internally mixed, so at a given time and location, all particles in a mode have identical fractional composition (as illustrated in Figure 1). As a result, most processes affect all aerosol species within a mode in an identical manner. E.g., if 5% (on a mass basis) of the Aitken mode particles coagulate with accumulation mode particles during a model time step, then 5% of the mass of each Aitken mode species is transferred to the corresponding accumulation mode species. Thus, extending MAM to treat the new MOA species involved few or no changes to process modules, and nearly all process modules automatically treat the new MOA species. The exception was emissions, where MOA-specific coding was added. Also, many processes utilize physical properties that are averaged or summed over all species in a mode (e.g., the total mass mixing ratio used in many processes, or the dry-volume-weighted hygroscopicity used in activation and water uptake processes), and the MOA species contribute to these.

MOA is initially emitted into either the accumulation or primary carbon mode (depending on initial mixing state assumptions; see below) and the Aitken mode. Aitken mode particles and their MOA are transferred to the accumulation mode by growth processes (condensation of $H_2SO_4$ and organic vapors and aqueous sulfate production) and coagulation. The transfer

by growth processes is termed renaming, wherein particles that grow larger than a size cut of ca. 80 nm diameter are transferred. Primary carbon mode particles and their MOA are also transferred to the accumulation mode by condensational growth and coagulation. The transfer due to condensation is termed aging, and particles that acquire a specified number of sulfate monolayers, or a hygroscopically equivalent amount of SOA, are transferred (Liu et al., 2012, 2016). The aging criterion used

was three monolayers, which resulted in an effective aging lifetime of approximately two days. The sensitivity of the model's aerosol lifetime to the aging criterion is discussed in Liu et al. (2016).

MOA and other aerosol species are also transferred to the coarse mode through evaporation of rain. When a rain drop completely evaporates, the aerosol material it contains (from in and below cloud scavenging that occurred at higher levels) is resuspended as a coarse mode particle. Because each rain drop is generally formed from thousands of cloud droplets, and the

10 CCN on which each cloud droplet formed, the resuspended particle is generally of coarse mode size (Wang et al., 2020). This differs from the earlier MAM treatment (Liu et al., 2012) in which particles resuspended from evaporating rain are returned to their original mode. This change mainly affects aerosol number concentrations (the number of particles resuspended is much smaller in the new treatment) and has a minor impact on aerosol mass concentrations.

## 2.6 Optical and cloud-forming properties of MOA

Particle CCN activation is determined by the Abdul-Razzak and Ghan scheme (Abdul-Razzak et al., 1998; Abdul-Razzak and Ghan, 2000; Ghan et al., 2011). The prescribed hygroscopicity for MOA is $\kappa_{\mathrm{MOA}} = 0.1$ (which is the hygroscopicity of, for example, xanthan gum, sometimes used as a proxy for marine organic matter; Dawson et al., 2016), compared with a sea salt hygroscopicity of $\kappa_{\mathrm{NCl}} = 1.16$. The prescribed density of MOA is 1,601 kg/m$^3$ (the density of alginic acid, a polysaccharide found within algal cell walls), compared with a sea salt density of 1,900 kg/m$^3$ (Table 1). The optical properties of MOA are

prescribed to be identical to those of sea salt aerosol, and are parameterized according to Ghan and Zaveri (2007). MOA did not contribute to ice nucleation in the current model configuration, but recent research indicates that marine organic particles can act as ice nucleating particles (INP), and may be an important source of INP to remote marine regions (Knopf et al., 2011; Burrows et al., 2013a; Wilson et al., 2015; DeMott et al., 2016; McCluskey et al., 2019; Zhao et al., 2021). Additionally, surfactant effects on aerosol activation (due to alteration of surface tension) are not treated, but evidence suggests that the

organic matter in marine aerosol is highly surface active (Blanchard, 1963; Barger and Garrett, 1970; Blanchard, 1975; Loglio et al., 1985; Giovannelli et al., 1988; Oppo et al., 1999; Mochida et al., 2002; Tervahattu et al., 2002; Cavalli et al., 2004; Facchini et al., 2008b), and that particle activation rates can be significantly modified for aerosol particles that contain salts mixed with substantial amounts of surfactants (e.g., 70% or more by mass) (Sorjamaa et al., 2004; McFiggans et al., 2006), suggesting a possible role for surface activity of marine organic matter in altering the water uptake and growth of marine

aerosol particles (Abdul-Razzak and Ghan, 2004; Ovadnevaite et al., 2011; Petters and Kreidenweis, 2013; Ruehl and Wilson, 2014; Ruehl et al., 2016; Dawson et al., 2016). It has been suggested that organic matter in submicron sea spray, by suppressing aerosol hygroscopic growth, may reduce the climate cooling associated with the scattering of sunlight by sea spray particles (direct aerosol effect; Randles et al., 2004).

**Table 3.** Simulation sensitivity cases

| Short name | Description |
| --- | --- |
| CNTL | Control experiment; no MOA |
| INT_ADD | Internal mixing; MOA adds to sea salt (default model) |
| INT_REPLACE | Internal mixing; MOA replaces sea salt |
| EXT_REPLACE | External mixing; MOA replaces sea salt |
| EXT_ADD | External mixing; MOA adds to sea salt |

## 3 Model simulations and analysis methods

Several sensitivity simulations were performed, which were identical in their configurations, except for changes in two model physical assumptions specific to MOA emissions, which are described in Section 3.1. All simulations were performed as free-running atmosphere-only climate simulations with year 2000 boundary conditions and fixed sea surface temperature. Ocean macromolecular concentrations, which drive the calculation of OMF in emitted aerosol, are provided to the model as climatological monthly mean values, and are the same in each year of the model simulation and across all sensitivity cases. The model was allowed to spin up for a full year in order to allow MOA concentrations (initialized at zero throughout the atmosphere) to fully equilibrate in the atmosphere. After the first year, ten additional years were simulated and all further analysis was performed using the climatological monthly means of the ten years.

### 3.1 Description of the control, default, and sensitivity cases

In implementing the emissions of MOA, decisions must be made about a number of factors, in particular (1) the mixing state of the aerosol, especially with respect to sea salt, with which it is co-emitted, and (2) the impact on the total number and mass of particles emitted. Experiments and observations currently do not provide precise constraints on how the mixing state and amount of emitted particles respond to different ocean biology and chemistry conditions. Therefore, we conducted sensitivity experiments with four sets of assumptions that bracket the extremes of possible responses.

An overview of the sensitivity cases tested is shown in Table 3. In the control simulation, no MOA is emitted. In the default treatment and three sensitivity cases, MOA is emitted using different assumptions in each case. In each case, we assume either fully "external" (EXT) or fully "internal" (INT) mixing, and we assume that marine organic emissions either REPLACE or ADD to the sea salt emissions that are natively simulated by the model. Of these four sensitivity cases, the INT_ADD case has been selected as the default for E3SM, and therefore will be given greater attention in our discussion of simulated aerosol and cloud impacts.

Next, we briefly summarize the experimental evidence regarding both assumptions, the implementation of these different sensitivity cases within the MAM4 modes, and the reasons INT_ADD was selected as the default case in E3SM.

### 3.1.1 Experimental evidence on sea spray mixing state response to ocean biology

The chemical mixing state of an aerosol population describes the extent to which individual particles contain multiple chemical constituents (internal mixing), as compared with particles composed of single chemical components that co-exist in a mixed population (external mixing; see schematic representation in Figure 1). The representation of mixing state in models can have important impacts on simulation of climate-relevant aerosol properties, including cloud condensation and ice nucleating particle concentrations, and aerosol optical properties (Riemer et al., 2019). In particular, for sea spray, we simulate a mixture of highly hygroscopic salts with organic matter that has low hygroscopicity. Therefore, the representation of mixing state can be expected to have important impacts on the simulation of cloud condensation nuclei, and it is important to both consider what is known about the mixing state of sea spray aerosol, and understand the extent of the model sensitivity to mixing state assumptions.

The experimental evidence that sheds the most light on the chemical mixing state of sea spray comes from artificial sea spray generation experiments, where sea spray particles can be measured immediately after emission, minimizing the potential for inclusion of secondary organic material from condensed gases. Experiments in which sea spray aerosol is generated by breaking waves in the presence of induced phytoplankton blooms provide the most realistic physical model of the sea spray aerosol production process (Ault et al., 2013; Collins et al., 2013; Prather et al., 2013). These experiments, combined with single-particle mass spectrometry and electron microscopy, have shown that the smallest emitted particles (up to about 100 nm in diameter) are primarily organic, and marine organic matter is typically mixed internally with sea salt upon emission for intermediate sizes (from about 200 nm to 1 µm in diameter). The vast majority of the largest particles (greater than 1.5 µm in diameter) are composed almost entirely of inorganic salts, with inorganic salts comprising about 20% of emitted particles with diameters close to 1 µm. Because the MAM4 accumulation mode extends from 80 nm to 1 µm, the majority of particles emitted in this size range are likely internally mixed, but some externally mixed particles (composed either of purely organic materials or purely inorganic salts) are also emitted in this size range. Therefore, while the internal mixing assumption is likely more consistent with current experimental evidence, it is also important to understand the impact of this simplifying assumption through the mixing state sensitivity cases.

### 3.1.2 Model implementation of sea spray chemical mixing state at emission

In MAM4, the chemical species within each aerosol mode are treated as internally mixed, an assumption that impacts the calculation of aerosol water uptake, activation, and optical properties. However, MAM4 represents two accumulation modes, which are externally mixed from each other: one of these is termed the "accumulation mode" and contains soluble aerosol species, while the other is termed the "primary carbon mode" and contains insoluble aerosol species. The insoluble components from the primary carbon mode are eventually transferred into the soluble accumulation mode due to aging, which in MAM4 is represented as occurring due to coating by condensation of volatile gases. Sea salt is always emitted into the soluble accumulation mode. In the "externally mixed" cases in this study, MOA is emitted into the primary carbon mode, where it is fully externally mixed with sea salt. In the "internally mixed" cases, MOA is emitted to the soluble accumulation mode, together with sea salt.

Because MAM4 has only one Aitken mode, MOA emissions in the Aitken mode are internally mixed with all other aerosol species in both cases.

### 3.1.3 Experimental evidence on sea spray number flux response to ocean biology

While mixing state is important, a larger impact of ocean biology on sea spray aerosol could potentially arise if ocean biology causes shifts in the total number and mass of emitted particles. However, there are fewer experiments that illuminate the impacts of ocean biological activity on the total number and mass of particles emitted, and they can be less straightforward to interpret. Perhaps the clearest experiment published to date that addresses this question may be from Alpert et al. (2015), which reported results from sea spray aerosol production in a phytoplankton mesocosm experiment using a plunging jet system for aerosol generation. They report an increase of sea spray aerosol particle number concentrations in the tank by a factor of about three when phytoplankton and bacteria were present in the tank, with the increase occurring mainly for particles less than 200 nm in diameter. While bubble generation was turned off, particle counts were the same with lights on and off, and the lamps used in the experiments put out photosynthetically active radiation with wavelengths of 400 to 700 nm (Alpert et al., 2015). Thus it is unlikely that the results are due to either SOA formation or the more recently recognized mechanism of UV-initiated (300–400 nm wavelengths) photosensitized reaction pathways at the air–water interface (Rossignol et al., 2016; Fu et al., 2015; Tinel et al., 2016; Bernard et al., 2016). A similar observation was made in an earlier study by Fuentes et al. (2011), where sea spray aerosol was artificially generated by a plunging multijet system, and aerosol emissions ($d < 200$ nm) increased substantially in the presence of phytoplankton exudates, with the magnitude of the increase varying depending on the phytoplankton species from which the exudate was derived.

Similarly, Long et al. (2014) also reported an increase in aerosol production in the presence of active biological production and light, in aerosol generation experiments using a plunging jet system to generate aerosol from natural seawater, onboard a ship. Increased aerosol production was observed only during daytime, and only in the biologically active waters of George's Bank (a coastal ecosystem); an increase was not observed in the oligotrophic waters of the Sargasso Sea. It is unclear, however, what mechanism caused the increased aerosol production in these experiments.

Finally, field observations have provided mixed evidence on the impacts of ocean biology on sea spray emission fluxes. In several cruises in the North Atlantic, sea spray aerosol was produced using a ship-board underway sea spray generator, and campaign-averaged sea spray flux and organic mass fractions were reported to show no seasonal differences (Bates et al., 2020). In contrast, Sellegri et al. (2021) reported that fluxes of sea spray and CCN, produced using a similar method were correlated with concentrations of ocean surface microbiota (nanophytoplankton cell abundances). Clearly, the source of these apparent discrepancies requires further investigation.

### 3.1.4 Model implementation of sea spray number flux response to MOA

To explore the model sensitivity to an assumed increase in sea spray emissions in response to ocean biology, we conducted pairs of sensitivity cases where organic matter is assumed to either REPLACE or ADD to the native emissions of sea salt aerosol. In REPLACE cases, the mass of emitted sea salt is reduced by an amount that is equal to the emitted MOA mass, such

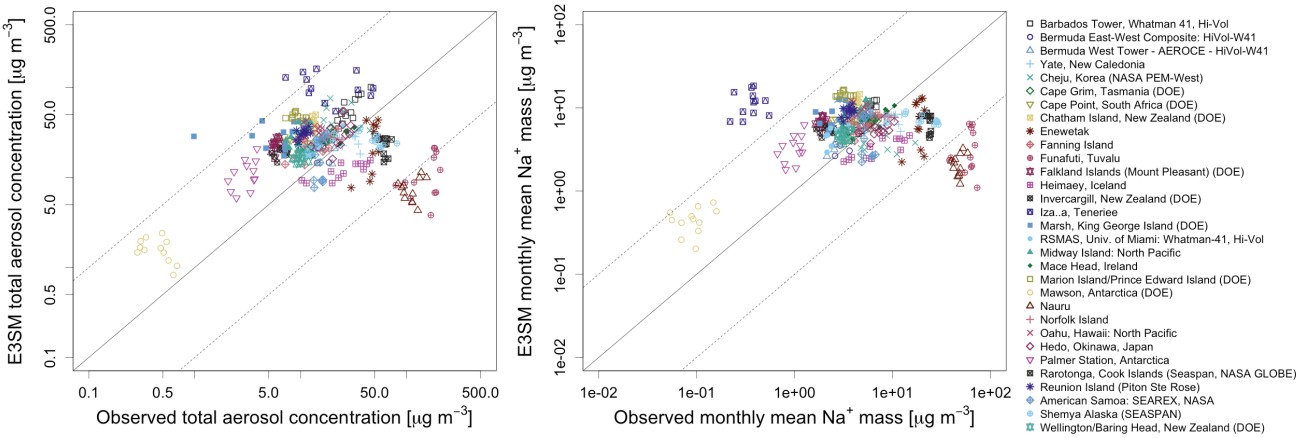

**Figure 2.** Comparison of monthly means of long-term observations at AEROCE/SEAREX stations with model climatological monthly means at nearest grid point. Left: Comparison of total aerosol concentration. Right: Comparison of sodium aerosol mass concentration. Because Na is well-conserved during transport compared to Cl and sea salt ions, sodium content of modeled sea salt aerosol has been approximated as 30.77% of modeled sea salt mass.

that the total emitted aerosol mass remains constant. The number of emitted sea salt particles is also reduced proportionally in each mode. If the underlying sea spray emissions parameterization is assumed to already include the organic content, then the REPLACE option would be the more physically plausible approach to implementing the OMF predicted by OCEANFILMS.

In contrast, if the underlying sea spray emissions parameterization is assumed to include only the inorganic salt components of the emitted spray, then the ADD option would be the more physically plausible approach. In the ADD cases, the mass and number of emitted sea salt are unchanged from the BASE model. Emitted MOA mass is added into the respective aerosol modes, increasing both the total mass and the total number of emissions in that mode.

Note that either the addition of MOA mass (ADD) or the replacement of sea salt by MOA (REPLACE) will impact the volume-weighted hygroscopicity of that mode, which is used in the droplet activation scheme (see Section 2.6 and Table 1).

## 3.2 Significance testing

The statistical significance of differences induced by the introduction of MOA emissions is presented for some key model fields in this paper. In each case, statistical significance of changes in a monthly or seasonal mean field was calculated by Welch's unequal variances t-test, treating the monthly or seasonal mean from each year of the ten-year simulation as an independent sample. The t-statistic was calculated in either each grid box of a 2-D field, or at each latitude after zonal averaging of a 2-D field.

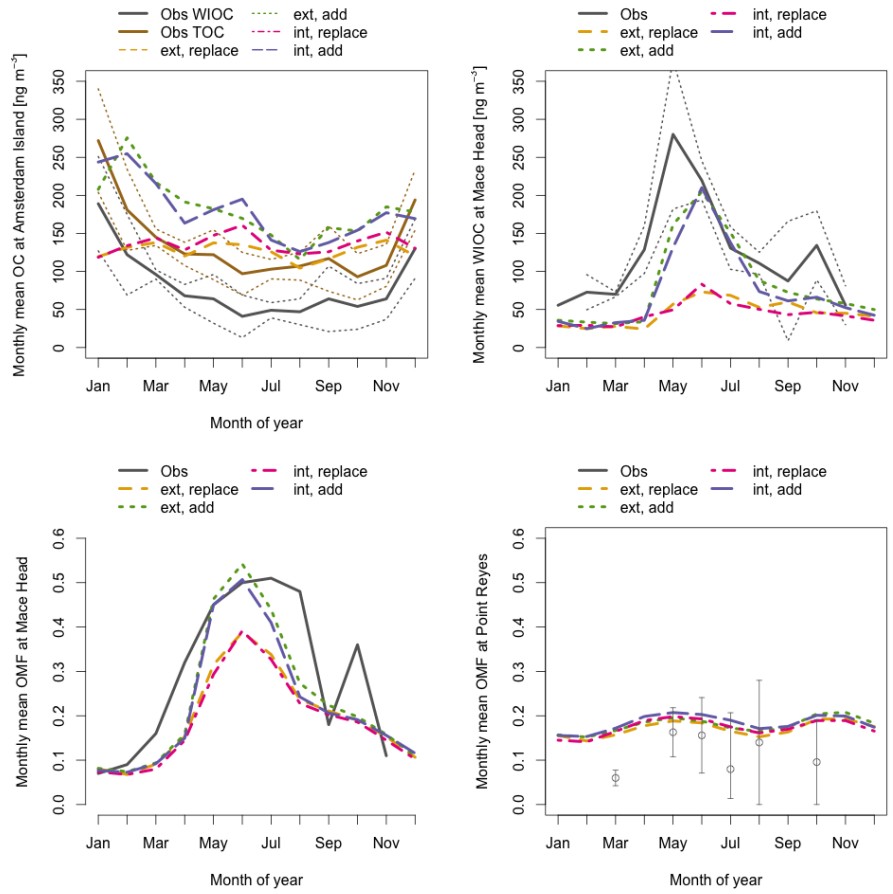

**Figure 3.** Top: Observed and simulated seasonal cycle, observed water-insoluble organic carbon (WIOC) and total organic carbon (TOC) in aerosol versus modeled marine organic carbon (MOC; converted from marine OM using OM:OC = 1.8), at Amsterdam Island (left: $d < 1.0$ μm; Sciare et al., 2009b) and Mace Head, Ireland (right: $d < 1.5$ μm; Rinaldi et al., 2013). Note that the Mace Head samples were selected for "clean marine" conditions as described in Rinaldi et al. (2013), and that pristine conditions typically prevail at Amsterdam Island. Dashed lines represent the standard deviation of measurements from the same month, where more than one observation was available for a given month. Bottom: Observed and simulated seasonal cycle, organic mass fraction of aerosol as reported under clean marine sampling conditions, Mace Head, Ireland ($d < 1.5$ μm; Rinaldi et al., 2013), and Point Reyes, California ($d <$ 2.5 μm; Gantt et al., 2011).

## 4 Model evaluation with observational data

While the overall characteristics of simulated aerosols in this model have been described in detail elsewhere, to provide context for this study, we present a brief observational comparison for simulated sea salt mass, which is particularly relevant to this study, followed by comparisons of MOA with available observational datasets. As with any comparison between a global

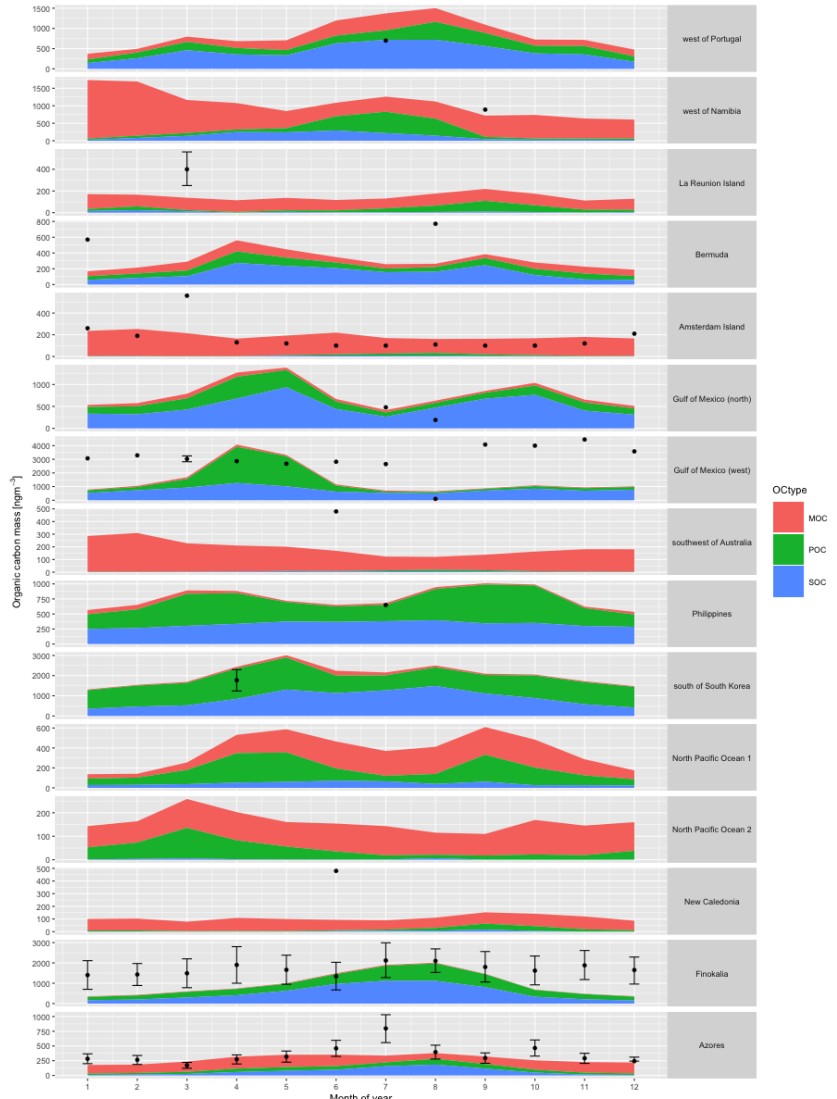

**Figure 4.** Seasonal cycle of simulated model total organic carbon (TOC) aerosol mass concentration at observational stations [ng/m³], resolved by organic aerosol source type. MOC: [primary] Marine Organic Carbon aerosol; POC: Primary [continental] Organic Carbon aerosol; SOC: [continental] Secondary Organic Carbon aerosol. Following Tsigaridis et al. (2013), the sum of the model's Aitken and accumulation mode organic aerosol mass is compared with PM2.5 OC observations (points) compiled in Bahadur et al. (2009), from marine observations originally published in Rau and Khalil (1993); Quinn et al. (2000); Ramanathan et al. (2001); Quinn et al. (2004); Bates et al. (2005); Quinn et al. (2006, 2008). Error bars, where shown, represent the standard deviations of measurements performed in a particular month. Point-to-point comparison of same observations is shown in Figure 5. Model OM values were converted to OC using the following OM:OC ratios: 1.8 (MOA:MOC), 1.9 (SOA:SOC), and 1.4 (POA:POC).

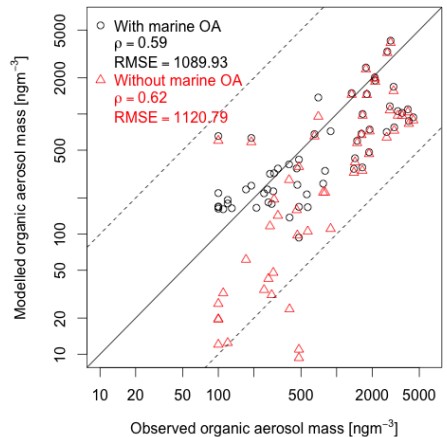

**Figure 5.** Observed versus simulated total organic aerosol (OC) mass concentration at observational stations [ng/m$^3$]; observations are the same as those shown in Figure 4. Black circles and red triangles are model climatological values (matched by month to observations and interpolated to observation location), from `INT_ADD` simulation, with and without MOA. Following Tsigaridis et al. (2013), the sum of Aitken and accumulation mode organic carbon mass is compared with PM2.5 OC observations compiled in Bahadur et al. (2009), from marine observations originally published in Rau and Khalil (1993); Quinn et al. (2000); Ramanathan et al. (2001); Quinn et al. (2004); Bates et al. (2005); Quinn et al. (2006, 2008). Model OM values were converted to OC using the following OM:OC ratios: 1.8 (MOA:MOC), 1.9 (SOA:SOC), and 1.4 (POA:POC). Pearson correlation coefficient and root-mean-square-error (RMSE) of model and observations are shown for model aerosol with and without marine organic contribution.

model and in situ observations, the model-observation agreement is limited in part by representativeness errors and the model's comparatively coarse resolution; i.e., observations will not always be representative for a model grid cell. In addition, field campaign data, which are affected by the weather and wind patterns of a particular time period, are being compared here with climatologies of monthly mean concentrations from the model, which is an imperfect comparison. Nevertheless, such
5  comparisons are critical to determine whether the model reproduces broad global geographic and seasonal patterns in observed concentrations over seasonal time scales, which are less susceptible to errors associated with the representativeness of short-term field campaign data.

## 4.1 Total and sea salt aerosol concentrations compared with in situ observations

Relatively few in situ observations are available that are appropriate for direct evaluation of sea spray aerosol on climatological
10  time scales; e.g., data from individual field campaigns may not capture seasonal and interannual variability. Therefore, to evaluate the overall simulation of sea spray aerosol, we compare with a benchmark dataset of in situ observational data collected by J. Prospero and colleagues at the University of Miami during the 1980s and 1990s, the AEROCE/SEAREX dataset. This dataset is freely available from the AEROCOM benchmark data website (http://aerocom.met.no/databenchmarks.html), and

includes filter measurements from a global network of marine stations, mostly located on islands. Most stations were located on windward shores or coasts, and filter samples were typically collected using a high volume sampler at 2 m above ground level, or mounted on a tower 10–20 m above ground level. At some sites, sampling was conducted only when winds were arriving from the site's marine sector; we did not emulate the sectored sampling, but we still include these sites in our comparison,

for consistency with previous studies (e.g., Tsigaridis et al., 2013). The aerosol chemical species measured at these stations typically included sodium, chloride, sulfate, nitrate, and methane sulphonic acid (MSA).

Comparisons between observed and simulated monthly mean climatological aerosol concentrations at the AEROCE/SEAREX stations are shown as scatterplots (Figure 2) for both sea salt and total aerosols in the default `INT_ADD` case. .The simulated sea salt burden is typically within a factor of ten of observations. Global model simulations of sea salt exhibit considerable

diversity (Textor et al., 2006a; Gliß et al., 2021), but variations of up to a factor of ten are typical for models of this class (e.g., Tsigaridis et al., 2013). However, we do note that the model falls outside of this range at a few locations, including the site at Izaña, Tenerife (yellow filled squares), where the model strongly overpredicts observed sea salt aerosol concentrations. Sea salt concentrations are underpredicted at Invercargill, New Zealand; Funafuti, Tuvalu; the Bermuda West tower, and sometimes at Miami. At Funafuti and Bermuda West, where aerosol concentrations are dominated by sea salt, the model also underpredicts

the total aerosol by a similar amount.

## 4.2    Evaluation of modeled MOA concentrations and organic mass fraction

We evaluate the simulated MOA concentrations and organic mass fractions using field observations of organic aerosol mass from station data, and from samples collected aboard ship campaigns.

Few observations are available that are appropriate for evaluating the simulated MOA at seasonal time scales in a global

model. To be useful for such an evaluation, observations must either (1) be obtained under conditions where organic aerosol mass is dominated primarily by the marine source, or (2) be capable of chemically distinguishing the primary MOA from secondary and non-marine aerosol sources. Further, the observation should ideally be obtained over a sufficiently long period of time to be presumed to be a representative sample, and should sample a sufficient portion of the seasonal cycle that responses to ocean biology can be observed. Very few datasets are available that meet these criteria. Among the existing datasets, different

studies have reported different observed variables. In this section, we describe and discuss the model-observation comparisons for three types of observational constraints on the seasonal cycle of sea spray organic matter – total organic carbon (TOC), water-insoluble organic carbon (WIOC), and organic mass fraction (OMF) of sea spray.

### 4.2.1    Comparison of marine OC and OMF seasonal cycles with site-based measurements under "clean marine" conditions

We focus first on an evaluation of the seasonal cycles of observed organic aerosol mass and OMF in the default `INT_ADD` model and the three sensitivity cases. For this evaluation, we focus narrowly on three coastal and island sites where long-term observations of organic aerosol are available that have either been screened for "clean marine" conditions (Mace Head, Ireland, and Point Reyes, California) or were collected in a region with minimal anthropogenic influence (Amsterdam Island, Southern

Ocean). In order to make these comparisons as physically meaningful as possible, in each case we compare only the quantities reported by the respective field experiment. At Amsterdam Island, observations were available of TOC and WIOC, while Mace Head observations include WIOC and OMF, and Point Reyes observations were available as OMF.

In our evaluation, we assume that under "clean marine" conditions, WIOC is attributable to primary sea spray organic matter (following Facchini et al., 2008b), and we therefore compare this variable directly with model-simulated MOA. We also focus particularly on the OMF as a metric for model evaluation, because this is the variable directly predicted by OCEANFILMS. While the prediction of TOC or MOA mass can be influenced by errors in other model processes (e.g., sea salt emissions, wet and dry removal rates, and emissions of other classes of organic aerosol for TOC), the prediction of OMF is only minimally influenced by model processes other than the OCEANFILMS partitioning of sea spray emissions. Therefore, measurements of OMF at seasonal scales and under "clean marine" conditions, provide the most direct test of the OCEANFILMS parameterization.

Comparisons with simulated seasonal cycles from all four model configurations are shown in Figure 3. In the upper left panel, we compare the model with observations from Amsterdam Island. Overall, the model's annual mean matches well with observed Amsterdam Island TOC in the `REPLACE` configurations (annual mean bias: $-3\%$, `EXT_REPLACE`; $4\%$, `INT_REPLACE`) and is biased high in the `ADD` configurations (annual mean bias: $39\%$, `EXT_ADD`; $37\%$, `INT_ADD`). However, positive correlations with the observed seasonal cycle are achieved only in the ADD configurations ($\rho = 0.55$, `EXT_ADD`; $0.68$, `INT_ADD`), while the REPLACE configurations have a seasonal cycle that is anti-correlated with observations ($\rho = -0.19$, `EXT_REPLACE`; $-0.47$, `INT_REPLACE`).

The upper right panel of Figure 3 compares the seasonal cycle of WIOC at Mace Head, Ireland. Observations at this site have been filtered for "clean marine" conditions as described in Rinaldi et al. (2013); we compare them with simulated MOA. At Mace Head, the ADD cases clearly match the observed seasonal cycle far better than the REPLACE cases, with both a lower annual bias in the annual mean ($-61\%$, `EXT_REPLACE`; $-62\%$, `INT_REPLACE`; $-29\%$, `EXT_ADD`; $-35\%$, `INT_ADD`) and a higher correlation ($\rho = 0.56$, `EXT_REPLACE`; $\rho = 0.67$, `INT_REPLACE`; $\rho = 0.81$, `EXT_ADD`; $\rho = 0.77$, `INT_ADD`). Again, the best correlation is achieved in the `INT_ADD` case.

The lower panels of Figure 3 compare the modeled and observed OMF at Mace Head (left) and at Point Reyes, California (right), where observations have also been screened for "clean marine conditions" in a similar fashion to those from Mace Head. Although the OMF of emissions is fixed, the `ADD` cases simulate higher OMF in boundary-layer aerosol at Mace Head; such a discrepancy can occur if aerosol from lower-OMF and higher-OMF regions mixes, due to the fact that the increases in total aerosol number and mass are disproportionately higher in the high-OMF regions for ADD cases. Once again, the `ADD` cases agree better with the Mace Head observations; at Point Reyes, all four configurations of the model give nearly identical results. Notably, the model reproduces the observed difference between a strong seasonal cycle at Mace Head and a weak or nonexistent seasonal cycle at Point Reyes. Gantt and Meskhidze (2013) accounted for this difference by introducing a dependence of the OMF of emitted aerosol on wind speed; however, Figure 3 shows that this difference in seasonal behavior at the two locations is also present in our simulations, in spite of the fact that OCEANFILMS does not assume a dependence of OMF on wind speed.

### 4.2.2 Comparison with observed TOC seasonal cycle at unscreened sites

For our default `INT_ADD` case, we performed additional comparisons with studies that have reported measurements of the seasonal cycles of total organic carbon (TOC) at coastal and island sites, but which did not attempt to screen for "clean marine" conditions. The advantage of these measurements is that their interpretation requires fewer assumptions, since it does not require us to assume that the "clean marine" screening procedures have adequately removed continental sources of organic aerosol. However, the lack of screening also requires us to compare with the total organic aerosol simulated by the model, and the comparison therefore becomes subject to model errors in other simulated organic aerosol components, including secondary organic aerosol and particulate organic carbon from burning of fossil fuels and biomass. Consequently, this measurement provides a strong benchmark for MOA sources only at times and locations where continental sources are very small, and errors in their simulation have negligible impact. Such conditions are frequent in the Southern Hemisphere remote oceans, but infrequent throughout most of the Northern Hemisphere (Hamilton et al., 2014).

We compare observed TOC seasonal cycles with the total organic carbon simulated by the model in the default `INT_ADD` case, from both continental and marine sources. Figure 4 shows the seasonal cycle of simulated monthly mean Aitken and accumulation mode organic carbon mass concentration (`INT_ADD`), subdivided into MOC, continental primary organic carbon (POC), and continental secondary organic carbon (SOC) mass concentrations, compared with climatologically averaged observed PM2.5 OC mass plotted as points. The model's organic aerosol mass has been converted to organic carbon (OC) mass using the OM:OC ratios given in the figure caption. Note that the model's PM2.5 also includes a portion of the coarse mode aerosol, which is not accounted for in this comparison. The observations used for this comparison are from the compilation by Bahadur et al. (2009), as shown in Tsigaridis et al. (2013). Note that both model and observational values represent climatological monthly means, where possible for the respective dataset. Model-observation agreement is greatly improved at sites where MOA dominates the OA mass: "Amsterdam Island", "west of Namibia", "La Reunion Island", "Bermuda", "southwest of Australia", and "New Caledonia". For most other sites, MOA contributes only a small fraction of the total OA, and has little impact on the total OA measurements.

Figure 5 shows the observed versus modeled monthly mean organic aerosol mass concentration for the same observations as shown in Figure 4. Only the mean value is compared for each station and month. Adding MOA improves model-observation agreement, improving the root-mean-squared error slightly from 1121 to 1090, although the correlation decreases slightly from 0.62 to 0.59. The small magnitude of the improvement in the objective RMSE metric obscures the fact that major improvements have been achieved in pristine locations where organic aerosol mass is small (Figure 4), but which therefore also have small impact on the RMSE. With inclusion of marine organic carbon, the simulated OC mass at all points falls within a factor of ten of the observations (i.e., between the dashed lines).

### 4.2.3 Comparison with findings from the NAAMES expedition

Another notable recent field experiment, the NAAMES expedition, observed sea spray flux and chemistry for sea spray generated shipboard from surface seawater (Bates et al., 2020). Given the unique marine aerosol observations collected by this

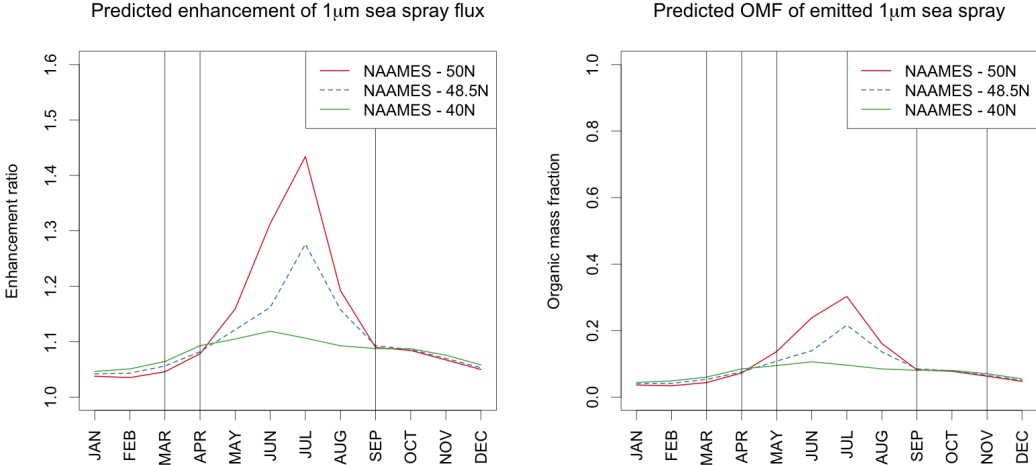

**Figure 6.** OCEANFILMS-predicted seasonal cycle of biological enhancement in the emission flux (left panel; ratio of predicted MOA+sea salt flux to flux of pure sea salt) and the organic mass fraction (right panel) of sea spray particles with $d < 1\mu$m, for the region of the NAAMES field campaign. The NAAMES campaign consisted of four cruises in different seasons, with most scientific sampling occurring close to 40 °W and between approximately 40 °N and 50 °N (Behrenfeld et al., 2019). All model results are monthly mean output at 40 °W, and at the latitudes indicated in the legend. Vertical lines indicate the months during which each variable was measured by NAAMES.

campaign, it is worthwhile to consider whether these data can be used to gain additional insight into the the behavior of OCEANFILMS.

First, we focus on the seasonal cycle of sea spray flux enhancement predicted by OCEANFILMS. Bates et al. (2020) reported that campaign-averaged sea spray number flux from NAAMES did not vary across four campaigns conducted during different seasons. At first glance, this result seems to contradict the assumptions and predictions of OCEANFILMS, which does predict seasonal cycles in these variables in this region. However, a closer examination is needed that accounts for the times and locations when sampling was conducted. Intensive scientific sampling in NAAMES occurred only during the months of April–May, September, and November (Behrenfeld et al., 2019). Figure 6 (left) shows the seasonal cycle of sea spray flux enhancement predicted by OCEANFILMS (in the default `INT_ADD` case) at three different latitudes in the NAAMES region, and indicates the months during which NAAMES measurements of sea spray flux were taken. NAAMES unfortunately was not able to measure sea spray number fluxes during those months when OCEANFILMS predicts a potentially detectable signal in this region, i.e., the months of May through August.

Similarly, NAAMES also reported no detectable difference between campaign-averaged OMF measured across four seasons (Bates et al., 2020). However, OCEANFILMS predicts almost no change in OMF this region (Figure 6, right), despite predicting enhanced OMF during June through August in the northern portion of the NAAMES study region.

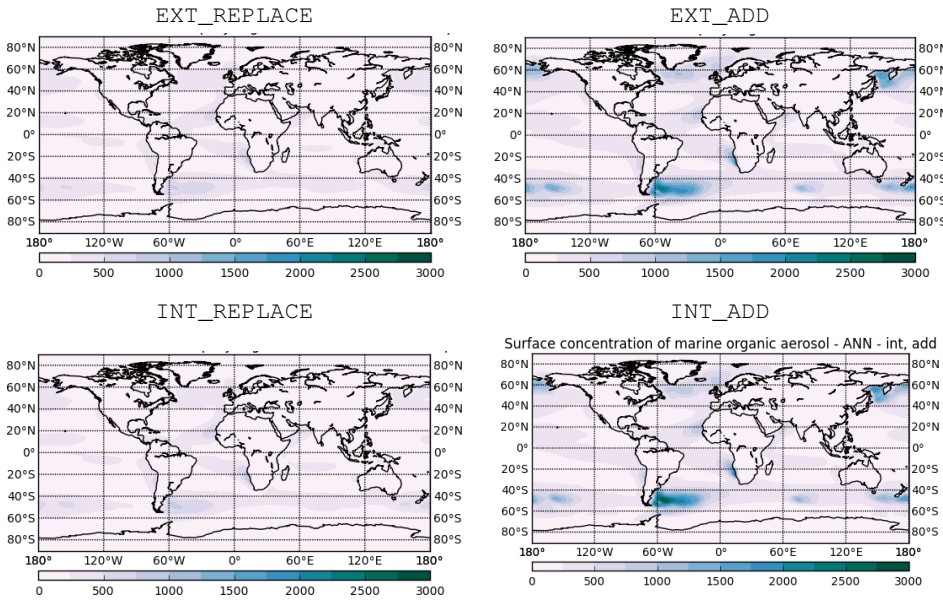

**Figure 7.** Simulated annual mean mass concentration [ng m$^{-3}$] of MOA (near-surface concentrations), for four sensitivity cases described in Table 3.

It is unclear whether weaker signals might have been present in aerosol OMF or flux that were not observable by NAAMES. With $N = 7$ or fewer filters analyzed for submicron OMF of generated aerosol per campaign, in the presence of high day-to-day variability (Lewis et al., 2021), NAAMES likely had sufficient statistical power to distinguish only extremely strong seasonal signals. Even the stronger organic enrichments that OCEANFILMS predicts would have occurred at latitudes around 50 °N during June and July (months during which NAAMES did not measure OMF or sea spray flux) might not be distinguishable from background variability with such a small number of samples.

In summary, our comparison with NAAMES data indicates that NAAMES did not measure at locations and times where OCEANFILMS predicts a strong signal in OMF or sea spray flux. Therefore, it appears that the lack of any seasonal differences in campaign-averaged OMF and sea spray flux reported by Bates et al. (2020) is nevertheless fully consistent with an OCEANFILMS implementation that does predict seasonal cycles in these variables.

## 4.3 Global MOA budgets and annual mean geographic distribution of MOA concentrations in the four sensitivity cases

Here we discuss the simulated MOA budgets and concentrations in the four sensitivity cases. Figure 7 shows the annual mean surface mass concentration [ng m$^{-3}$] of MOA in all sensitivity cases. Annual mean mass concentrations are $< 250$ ng m$^{-3}$ over much of the globe in all sensitivity cases. However, in the ADD cases, much higher annual mean concentrations are produced, which can exceed 2.5 µg m$^{-3}$ locally. These higher concentrations are the result of higher emissions in the ADD cases; since

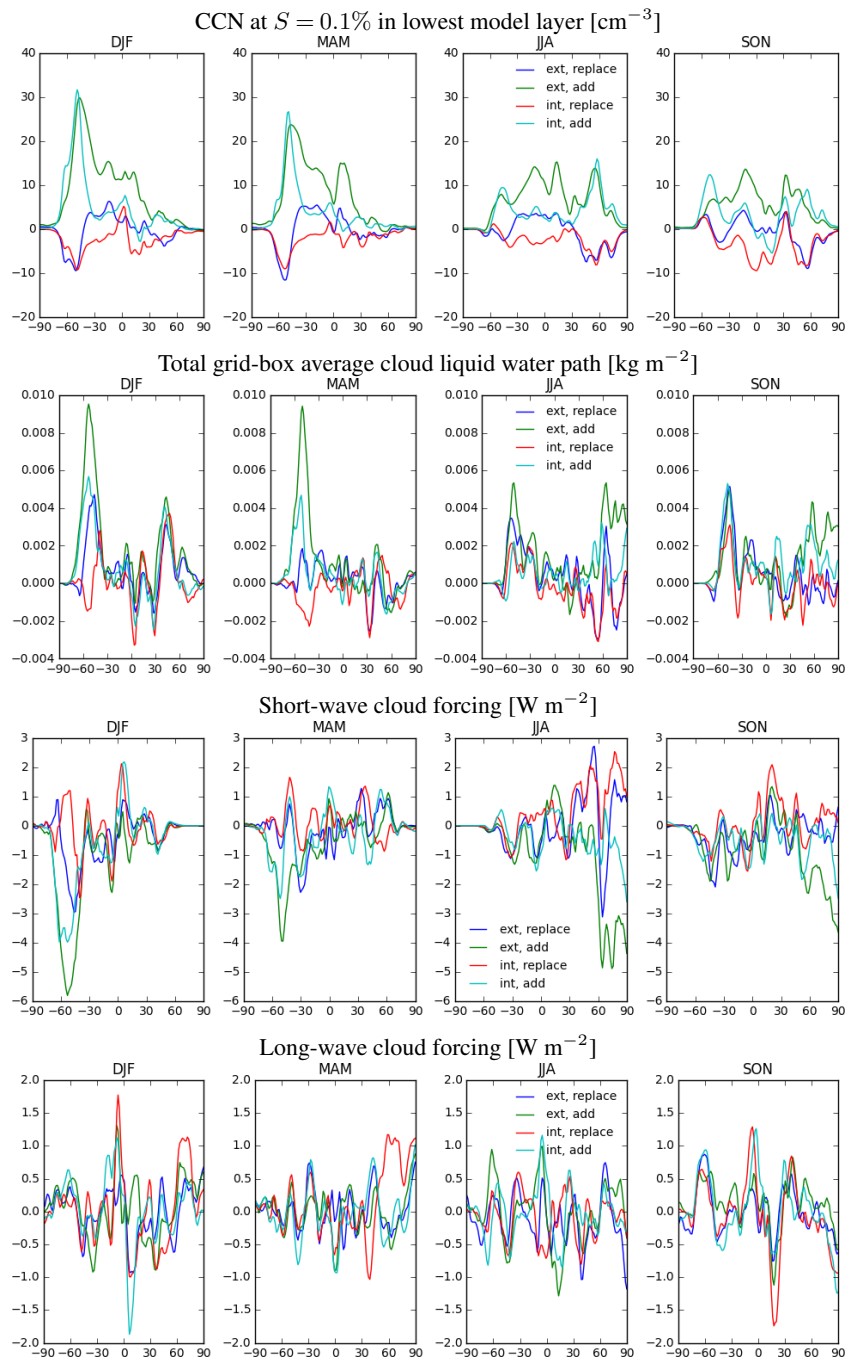

**Figure 8.** Difference in zonal mean for marine organics cases, relative to control simulation, by season. Top: CCN at $S = 0.1\%$ in lowest model layer [cm$^{-3}$]. Second row: Total grid-box average cloud liquid water path [kg m$^{-2}$]. Third row: Short-wave cloud forcing [W m$^{-2}$]. Fourth row: Long-wave cloud forcing [W m$^{-2}$].

**Table 4.** Global atmospheric burdens, source terms, and residence times for MOA in the Aitken mode, and the combined accumulation and Primary carbon modes, for each sensitivity case. Residence times are calculated as the global annual mean burden divided by total global annual losses due to wet and dry deposition.

| | MOA global metrics | | | | |
| --- | --- | --- | --- | --- | --- |
| | | external | | internal | |
| | | replace | add | replace | add |
| Aitken | Global annual mean burden (Gg) | 0.05 | 0.09 | 0.05 | 0.09 |
| | Global annual source (Gg/yr) | 63.5 | 96.3 | 64.2 | 95.7 |
| | Global annual mean residence time (d) | 0.30 | 0.33 | 0.29 | 0.33 |
| Accumulation + Primary carbon | Global annual mean burden (Gg) | 31.3 | 44.3 | 34.3 | 47.8 |
| | Global annual source (Gg/yr) | $7.71 \times 10^3$ | $11.3 \times 10^3$ | $10.6 \times 10^3$ | $14.6 \times 10^3$ |
| | Global annual mean residence time (d) | 1.1 | 1.1 | 1.2 | 1.2 |

OMF of emitted aerosol is approximately the same in both cases, the emissions of MOA required to achieve a given value of OMF are much higher in the ADD case than in the REPLACE case. Global mean concentrations are slightly higher in the externally mixed cases, a reflection of the longer atmospheric residence time of aerosol emitted into the externally mixed primary carbon mode, until it is aged into the accumulation mode.

Table 4 shows the global atmospheric burdens, source terms, and residence times, with respect to all removal processes, for MOA in the Aitken, accumulation and primary carbon modes, in each sensitivity case. Note that the source term is significantly higher for the ADD cases than for the REPLACE cases because the total sea spray emissions are modified in the ADD case while holding the OMF of those emissions constant. Differences between the source terms in the internal and external mixing cases arise from the use of the primary carbon mode to accomplish external mixing with the accumulation mode.

The mean global residence time ranges from 0.29—0.33 days for the Aitken mode, and 1.1—1.2 days for the accumulation and primary carbon modes, with small variations across sensitivity cases. In the Aitken mode, losses to dry deposition predominate, while losses in the accumulation mode are primarily due to wet deposition. These lifetimes are much shorter than those computed for continental emissions of aerosols such as dust in similar models Wu et al. (2020), which is consistent with expectations for sea spray aerosol and for marine-emitted aerosols in general (Textor et al., 2006b; Burrows et al., 2013b). Most global climate models produce frequent light rain in marine regions, which strongly controls the rate of aerosol wet removal over oceans, and which may produce biases in the geographic distribution of simulated aerosol burden, especially over oceans (Wang et al., 2021). Recent work by Emerson et al. (2020) also suggests that the dry deposition parameterization used in E3SM may overestimate the rate of dry deposition to water surfaces in the Aitken and accumulation modes. However, since the simulated effects on CCN and clouds are controlled mostly by the accumulation mode, model biases in dry deposition have only minor effects on the main results of this study.

## 4.4 Zonal mean responses of cloud condensation nuclei and cloud variables in the four sensitivity cases

Next we examine the model-simulated CCN and cloud responses in the four sensitivity cases. Figure 8 shows the seasonal zonal mean changes in several relevant model fields, for all four sensitivity cases. As the number of total aerosol particles increases in the `ADD` cases, this leads directly to increases in boundary-layer CCN concentrations (at supersaturation $S = 0.1\%$; Figure 8, top row). This is particularly true in the austral summertime (DJF) over the Southern Ocean, where the boundary-layer CCN ($S = 0.1\%$) number increases by up to 30 cm$^{-3}$, and by 5–10 cm$^{-3}$ in winter, thus adding about 20–25 cm$^{-3}$ to the seasonal cycle in CCN number. For context, Ayers and Gras (1991) reported a summer–winter difference of 50 cm$^{-3}$ in CCN ($S = 0.23\%$) in a nine-year observational record at Amsterdam Island. McCoy et al. (2015) reported an average summer–winter difference of 10 cm$^{-3}$ in satellite-observed cloud-top CDNCs over oceans from 35 S to 55 S, and attributed the seasonal cycle primarily to sulfate aerosol and secondarily to organic matter in sea spray aerosol. In the `REPLACE` cases, by contrast, CCN numbers are depressed over the Southern Ocean and Arctic in summertime, which is inconsistent with observational evidence. This is an expected feature of the simulation in the `REPLACE` cases, since salt emissions are reduced and replaced by less-hygroscopic organic matter.

In response to the changes in CCN number, there are also changes in simulated CDNC and cloud liquid water path (LWP; Figure 8, second row). In particular, the zonal mean values of CDNC and LWP both increase over the summertime Southern Ocean in the `ADD` and `EXT` cases. Zonal-mean LWP also increases in the summertime Arctic in the `EXT_ADD` case, but decreases in the `REPLACE` cases. The changes in these two variables were clearly distinguishable from natural variability over the Southern Ocean ($p < 0.1\%$), while they were largely not distinguishable from natural variability over the Arctic (t-test calculations for significance not shown). The stronger cloud responses over the Southern Ocean (as compared to the Arctic) are likely attributable largely to two related causes. First, the presence of significant landmass in the Northern Hemisphere means that the oceanic area where changes in sea spray emissions can take effect is significantly smaller. Second, the emissions of sea spray organic matter are smaller in the Arctic relative to the Southern Ocean (Figure 7), resulting in smaller changes in zonal mean CCN (Figure 8, top row). Finally, it is possible that cloud sensitivity to changes in CCN concentrations differs between the Arctic and the Southern Ocean.

These changes in CDNCs and LWP lead to changes in cloud radiative forcing. Zonal mean short-wave cloud forcing changes (Figure 8, third row) are strong, and significantly different from the natural variability ($p < 0.1\%$), over the summertime Southern Ocean in the `INT_ADD` and `EXT_ADD` cases, and over the summertime Arctic in the `EXT_ADD` case. Changes in long-wave cloud forcing (LWCF), by contrast, are smaller (Figure 8, fourth row) and do not differ significantly from natural variability (t-test calculations for significance not shown).

## 4.5 Selection of `INT_ADD` as the default case for E3SM

As previously noted, experimental evidence does not provide fully definitive guidance on which of the four sensitivity cases is most realistic (Section 3.1).

However, after considering the balance of experimental evidence, in combination with the evaluation of simulated aerosol fields described herein, we have selected the `INT_ADD` case as the model default.

Our reasons for selecting an `ADD` case as the default are fourfold:

1. **Consistency with observed seasonal cycles:** The `ADD` cases appear to produce better simulations of the MOA seasonal cycles, for the small number of available observations of that are appropriate for such a comparison (Section 3.1).

2. **Bottom-up evidence from laboratory experiments:** Several recent laboratory experiments have shown that artificially generated sea spray production can increase in the presence of in the presence of artificially induced phytoplankton blooms. These include experiments by Fuentes et al. (2011) and Alpert et al. (2015), both showing an increase in aerosol number emitted ($d < 200$ nm), and by Forestieri et al. (2018), showing an increase in total particles emitted.

3. **Top-down evidence from satellite observations:** McCoy et al. (2015) showed from top-down satellite constraints that cloud drop number concentrations are elevated above phytoplankton blooms, suggesting an increased number of CCN in those regions that could not be explained by modeled sulfate or sea salt aerosol.

4. **Bottom-up evidence from field experiments:** Finally, we note that two recent field studies have attempted to detect biologically driven signals in sea spray number flux using underway seawater plunging jet systems onboard ships that were fed with ocean surface waters, which both appear to be consistent with the `ADD` assumption in OCEANFILMS. Sellegri et al. (2021) measured CCN number fluxes in three ocean basins, and found that increases in both CCN emissions and number flux of sea spray aerosol ($d > 100$ nm) were correlated with the presence of certain groups of organic molecules and microorganisms in surface seawater. This implies an influence of marine biology on number and mass emission fluxes, which is qualitatively consistent with the `ADD` assumption. The second study, NAAMES, detected no seasonal trends, but measured only in places and times where OCEANFILMS also predicts negligible seasonal trends (see Section 4.2.3). Therefore, the findings from this NAAMES appear to be fully consistent with either the `ADD` or the `REPLACE` assumption.

The primary reason for selecting an INT case is the experimental evidence indicating that at least some degree of internal mixing occurs during the emission of sea spray organic matter, particularly in the accumulation mode, which accounts for most CCN (Prather et al., 2013). It is also worth noting that the results for the `INT_ADD` and `EXT_ADD` cases are relatively similar, so the choice between internal and external mixing is less consequential than the choice to add to (rather than replace) emitted mass.

For these reasons, we have selected the `INT_ADD` case as the default for E3SM, and our remaining analysis focuses on this configuration of the model.

## 5 Impacts of OCEANFILMS on aerosol and clouds in E3SM's default `INT_ADD` configuration

We now turn to a more detailed discussion of impacts of OCEANFILMS on E3SM aerosol and clouds in the model's default `INT_ADD` configuration.

**Table 5.** Annual and DJF mean relative (percentage) changes in aerosols, clouds, and radiative fluxes, globally and over the Southern Ocean (`INT_ADD` case).

| | Mean change (%) from control simulation, `INT_ADD` simulation | | | | |
|---|---|---|---|---|---|
| Study | Gantt et al. (2012) | This study | | | |
| Variable | Annual | Annual | DJF | Annual | DJF |
| | global | global | global | 20–90 S | 20–90 S |
| CCN, boundary layer ($S = 0.1\%$) | | 3.66 | 4.66 | 13.01 | 22.93 |
| Cloud LWP (g m$^{-2}$) (Grid average) | 0.21 | 7.0 | 6.0 | 2.1 | 2.4 |
| SWCF (W m$^{-2}$) | $-0.12$ | $-0.36$ | $-0.4$ | $-0.74$ | $-1.64$ |
| LWCF (W m$^{-2}$) | | $-0.01$ | $-0.08$ | $-0.20$ | $-0.27$ |

## 5.1 Changes in global distribution of aerosol emissions, chemistry, and amount between `INT_ADD` and `CNTL`

Introducing the MOA representation (`INT_ADD` configuration) directly impacts aerosol chemistry, aerosol number emissions and concentration, and CCN number, particularly over biologically active marine regions. The top panels of Figure 9 show the annual mean organic mass fraction in near-surface air and at 850 hPa. Over emission "hot spots" such as southeast of South
America, OMF is slightly lower at 850 hPa, relative to the near-surface air, likely due to mixing with lower-OMF aerosol from local sources. However, overall the OMF at 850 hPa is quite similar to OMF in the model's surface layer, indicating that sea spray organic matter is transported with sea salt to altitudes relevant for cloud formation.

The second row of Figure 9 shows the absolute and relative annual mean changes in annual mean accumulation mode aerosol number emissions. Statistically significant increases occur in accumulation mode number emissions over much of the Southern
Ocean and Arctic, with annual mean emissions more than doubling in some regions. The impact of these increased emissions on accumulation mode number concentration in near-surface air is shown in the third row of Figure 9. Annual mean number concentrations approximately double over the strong phytoplankton blooms off the southeastern coast of South America, and smaller, but still significant changes occur in the northern Hemisphere around Greenland and the Bering Strait.

## 5.2 Changes in CCN, cloud properties, and radiative fluxes in `INT_ADD`

### 5.2.1 Southern Ocean

These increases in aerosol number translate into increases in CCN ($S = 1\%$) number concentration (Figure 9, bottom row), with annual mean increases exceeding 50 cm$^{-3}$ regionally. Because the responses of CCN to aerosol number, and of CDNC (Nd) to CCN are approximately logarithmic, it is especially helpful to examine relative changes in aerosol number and CCN

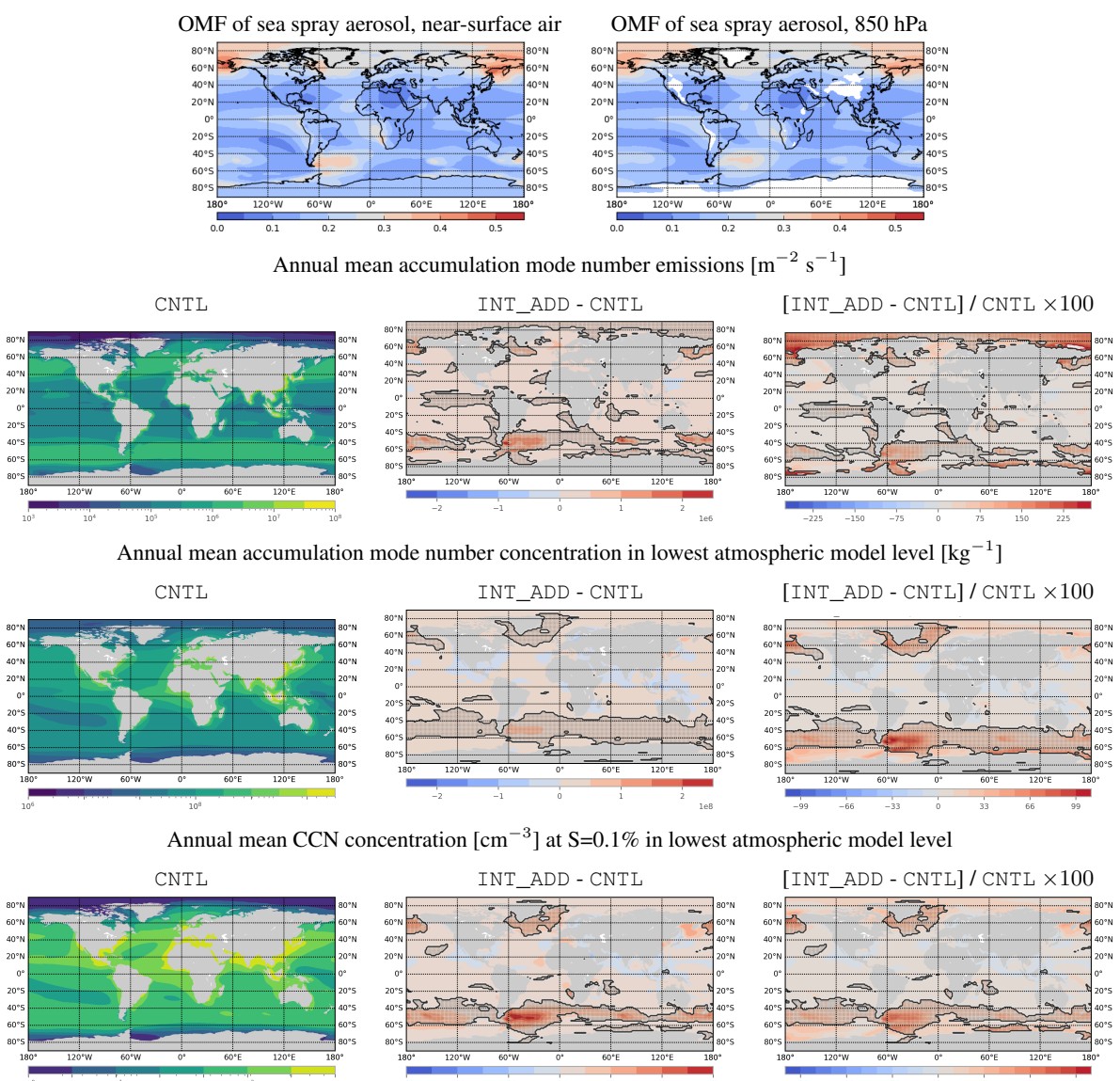

**Figure 9.** Annual mean modelled fields, and changes in annual mean modeled fields due to introduction of MOA in the `INT_ADD` case. Shaded regions outlined with black contours indicate regions where absolute differences are statistically significant by Welch's t-test at the $p < 0.1\%$ level. White space indicates missing values; continents are shaded grey in the second through fourth rows. Top row: Global annual mean organic mass fraction of accumulation-mode sea spray aerosol at surface (left) and 850 hPa (right). Second row: Annual mean accumulation mode number emissions [m$^{-2}$ s$^{-1}$] in `CNTL`(left), and absolute change (center) and percentage change (right) in `INT_ADD`. Third row: As above, for annual mean accumulation mode number concentration in lowest atmospheric model level [kg$^{-1}$]. Fourth row: As above, for annual mean CCN concentration [cm$^{-3}$] at S=0.1% in lowest atmospheric model level.

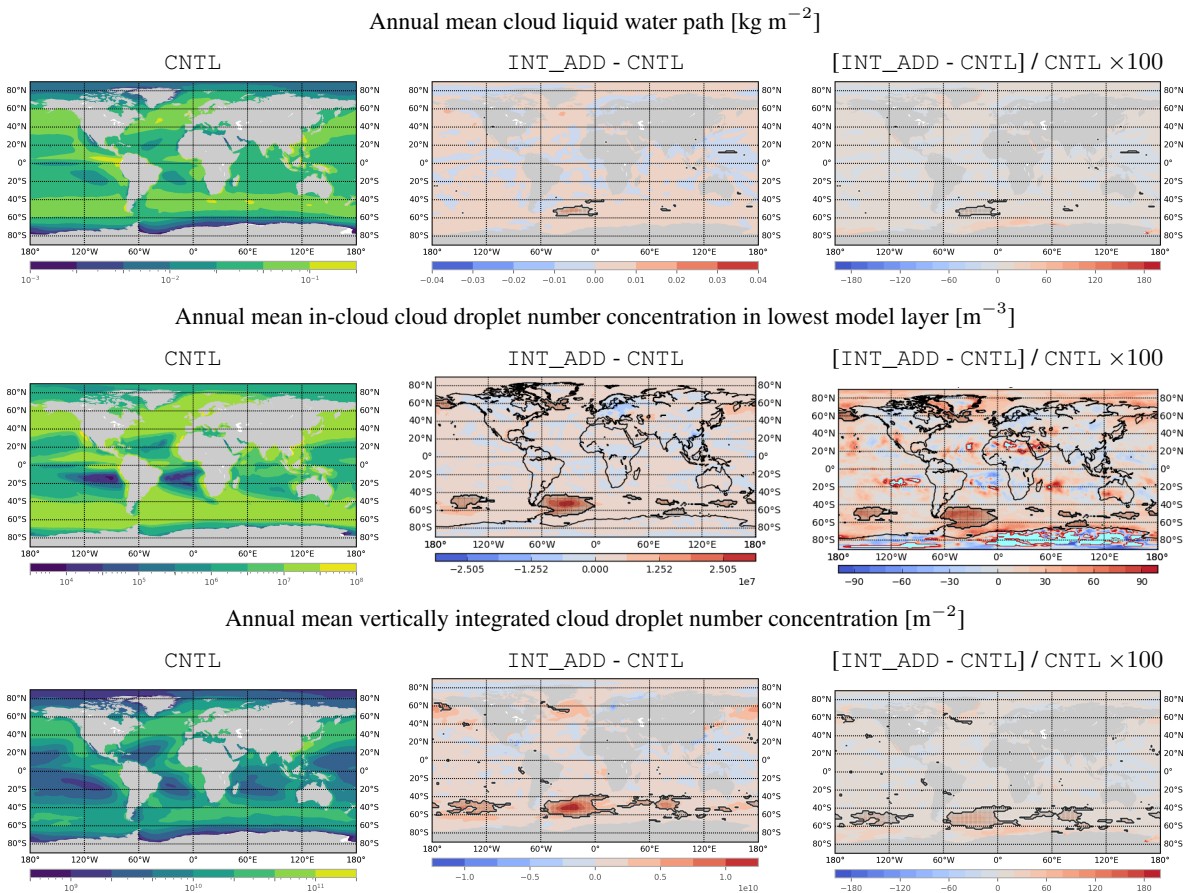

**Figure 10.** Annual mean modeled fields, and changes in annual mean modeled fields due to introduction of MOA in the `INT_ADD` case. Shaded regions outlined with black contours indicate regions where absolute differences are statistically significant by Welch's t-test at the $p < 0.1\%$ level. White space indicates missing values. Top row: Annual mean cloud liquid water path [kg m$^{-2}$] in `CNTL` (left), and absolute change (center) and percentage change (right) in `INT_ADD`. Second row: As above, for annual mean in-cloud cloud droplet number concentration in lowest model layer [m$^{-3}$]. Third row: As above, for annual mean vertically-integrated cloud droplet number concentration [m$^{-2}$].

(Figure 9, bottom right panel). Relative increases of 20%–50% in annual mean CCN ($S = 0.1\%$) concentration occur over much of the ocean from $40°$S to $60°$S.

Absolute changes in annual mean areal cloud fraction for low, mid-level, and high clouds do not exceed 6% at any location, and local differences in cloud areal fraction mostly do not pass criteria for statistical significance (not shown). This implies that
5 aerosol impacts on clouds are mostly related to increases in annual mean cloud thickness (as indicated by cloud liquid water (Figure 10, top row) and annual mean in-cloud CDNC (Figure 10, second row), which combine to produce relative changes in vertically integrated cloud droplet number concentration exceeding 20% over much of the Southern Ocean (Figure 10, third

row). Note that changes in these annual-mean cloud variables can be caused by either changes in the properties of individual clouds, or changes in the temporal behavior of clouds, i.e., frequency of cloud formation and occurrence, and cloud lifetime, or a combination of both.

In an earlier study to which some of us contributed (McCoy et al., 2015), we analyzed spatial and regional patterns in satellite-observed cloud-top droplet number concentrations over the Southern Ocean ($35°$ to $55°$ S) in order to infer the contribution of CCN from MOA to increases in total CDNC, and hence, to radiative forcing via the cloud albedo effect. That study concluded that increases in cloud albedo due to MOA contributed approximately 1–2 W/m$^2$ additional annual mean radiative cooling over the Southern Ocean, with the cooling occurring primarily during the austral summer months (DJF).

In our simulations, zonal annual mean shortwave cloud forcing (SWCF) in the Southern Ocean is strengthened by an average of $-1.261$ W/m$^2$ and about $-4$ W/m$^2$ around $50°$S in austral summer in the `INT_ADD` case, with the negative sign indicating strengthened cooling (Table 5; Figure 8). As shown in Table 5, this is about three times the global annual SWCF simulated by Gantt et al. (2012), where the aerosol indirect effect was also estimated to have decreased by up to 0.09 W m$^{-2}$ (7%). To better understand the robustness of this finding, we additionally tested for statistical significance of the changes in annual mean and seasonal mean SWCF using the Welch's t-test statistic. When examining the statistical significance of changes on a grid-cell basis, no model grid cells show local statistically significant changes in the annual mean, while a small region to the east of Cape Horn shows local changes that have statistical significance; this is also the region where the model simulates the strongest statistically significant impacts on cloud drop number concentration, LWP, and ice water path (Figure 10).

To place these findings in context, it is important to note that Figure 8 and Figure 10 attempt to evaluate the statistical significance of a comparatively small signal in the presence of significant noise at the grid-box level. It is a well-established and characteristic behavior of free-running atmospheric model simulations that local perturbations will rapidly produce large differences in the model's physical state due to chaotic dynamical responses, producing noisy model responses (Lorenz, 1963; Wan et al., 2017). This noise can obscure the signal of simulated impacts in the model, unless that signal is locally very strong. One strategy for overcoming this signal-to-noise issue is to perform averaging over larger regions, e.g., to determine whether a robust signal can be detected in the zonal or average mean. This is analogous to the familiar "signal averaging" approach used in many experimental fields, which takes advantage of the fact that when averaging across many samples (in this case, model grid boxes), a physical signal will tend to accumulate, while random noise tends to be reduced by the averaging process (Hassan and Anwar, 2010).

We therefore explore the cloud impacts further for the `INT_ADD` case by examining the changes in zonal mean cloud cover for statistical significance. As shown in Figure 11, we find that over the Southern Ocean, despite the model's internal variability, the simulation produces robust zonal mean changes in SWCF over the Southern Ocean that are significant at the $p < 0.5\%$ level during all seasons, and at the $p < 0.1\%$ level during DJF and autumn (MAM). This is a key finding of the present study, and is consistent with the top-down constraints on changes in zonal mean SWCF from McCoy et al. (2015), lending increased confidence that both the model result and the values inferred from the satellite observations are realistic.

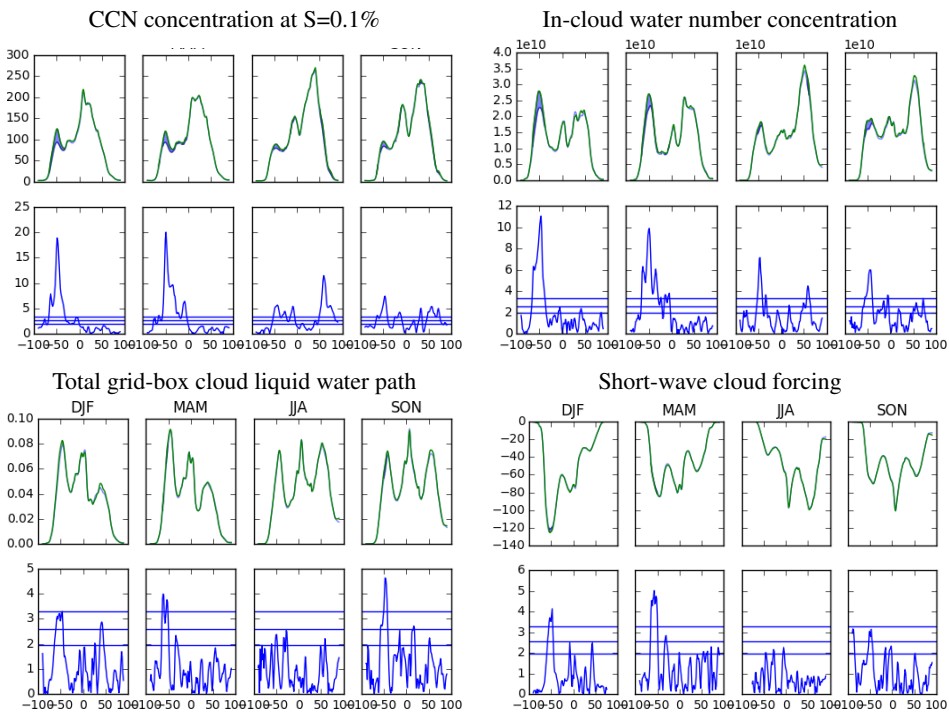

**Figure 11.** Zonal mean fields from control (black) and `INT_ADD` sensitivity case (green) for each season, and Welch's t-test statistic. Shading in top panels highlights regions where differences are significant at the $p < 0.1\%$ level, and horizontal lines on bottom panels indicate levels of significance at $p < 0.1\%$, $p < 1\%$, and $p < 5\%$. Variables shown are: CCN concentration at S=0.1% [top left]; in-cloud water number concentration [top right]; total grid-box cloud LWP [bottom left]; short-wave cloud forcing [bottom right].

### 5.2.2 North Atlantic, North Pacific and Arctic

Strong responses to the new MOA representation are also observed in the Arctic in some sensitivity cases, with the organic mass fraction exceeding 30% throughout most of the Arctic. While observations of aerosol and clouds from the Arctic are limited due to the difficulty of performing such observations, those observations that do exist suggest that in the central high Arctic, biogenic primary marine aerosol may contribute a large fraction of the aerosol particles and CCN (Bigg and Leck, 2001; Leck et al., 2002; Lohmann and Leck, 2005; Leck and Bigg, 2005a; Orellana et al., 2011). The model responds to these emissions of marine organic matter in the Arctic with a strengthening of zonal mean SWCF in the `EXT_ADD` case, and weaker responses in SWCF in the other sensitivity cases (Figure 8 see also previous discussion in Section 4.4).

In contrast with the Southern Ocean, inputs of MOA in the North Atlantic and North Pacific do not have appreciable impacts on the clouds there. This is likely because significant changes in clouds arise where background aerosol concentrations are low, and additional aerosol therefore produces large relative changes in CCN number concentrations (Pringle et al., 2012; Carslaw et al., 2013). Changes in aerosol and CCN concentration in the North Atlantic are statistically significant and of similar absolute

magnitude to changes in the Southern Ocean, but in the Northern Hemisphere, these MOA inputs sources are small relative to the continental aerosol sources that dominate the aerosol population in this region. In the Southern Hemisphere, where continental sources are minimal, the impact is much larger.

## 6   Summary, conclusions and outlook

Experimental findings during recent decades have firmly established that submicron sea spray aerosol contains significant contributions of MOA that is linked to ocean biology. Here we have described the implementation of OCEANFILMS, a mechanistic parameterization of MOA emissions (Burrows et al., 2014), in the E3SM global climate model, and the simulated responses of aerosols, clouds, and the Earth's energy balance in atmosphere-only simulations with fixed sea surface temperature. Experimental evidence does not yet provide detailed and conclusive guidance on some aspects of the process, such as the extent to which sea spray organic matter is internally mixed with salt, and the extent to which biological activity alters the total sea spray flux. With this in mind, four simulations were conducted to explore the sensitivity of simulated aerosols and clouds to the assumptions about MOA mixing state and impacts of MOA on total sea spray flux. Overall, we find that the mixing state assumption has relatively small impact on the simulation, while larger differences are observed between those cases where sea spray organic matter either ADDs to or REPLACEs sea salt. The default `INT_ADD` case and the `EXT_ADD` sensitivity case produce generally similar results, and both produce a reasonable prediction of the observed seasonal cycles at three sites with seasonal-scale observations under clean marine conditions. This indicates that the OCEANFILMS model captures the main features of the observed seasonality in OMF and MOA at these locations, including the observed differences between the sites. In particular, OCEANFILMS reproduces the contrast between a strong seasonal cycle at Mace Head, Ireland, and a weak seasonal cycle in OMF at Point Reyes, California. In a previous study, Gantt and Meskhidze (2013) report that their Chl-$a$-based parameterization initially did not reproduce this geographic difference, and they therefore introduced an assumed dependence of OMF on surface wind speeds in order to match the observations. In contrast, OCEANFILMS does not require any additional assumptions to reproduce this geographic difference, because the observed difference between these two sites is already captured by the underlying ocean biogeochemistry simulation.

Of the four sensitivity cases explored, the `INT_ADD` case was selected as the default for the E3SM model. This case, which assumes that MOA is externally mixed with sea salt and adds to the natively simulated sea salt emissions, produces good agreement with observations of the seasonal cycles of OMF and MOA under clean marine conditions. Its assumption that organics are mixed internally with salts in sea spray is qualitatively consistent with recent experimental evidence (Prather et al., 2013). Its assumption that marine biological activity can be associated with increases in the emitted sea spray number and mass is also qualitatively consistent with several recent laboratory and field experiments (Fuentes et al., 2010; Alpert et al., 2015; Forestieri et al., 2018; Sellegri et al., 2021). Finally, simulated cloud impacts in the `INT_ADD` configuration are consistent with indirect evidence from an analysis of satellite observations (McCoy et al., 2015). For these reasons, the `INT_ADD` case has been selected as the default for the E3SM model. These assumptions, and other aspects of the parameterization, could potentially be refined in the future as additional experimental evidence emerges.Additional work is also needed to compare the

OCEANFILMS parameterization with previous, [Chl-*a*]-based parameterizations of the emitted OMF and with the constant OMF hypothesis advanced by Quinn et al. (2014).

Our comparison with existing observations shows that MOA simulated by OCEANFILMS agrees relatively well with the few existing datasets that provide useful constraints on the simulated seasonal cycles, but also reveals how few observations are available that provide such a constraint. Our comparison with the observations from the recent NAAMES field campaign in the North Atlantic, for example, demonstrates the critical importance of the campaign's spatiotemporal sampling strategy. In particular, although OCEANFILMS does predict a strong seasonal cycle in OMF and sea spray flux in the North Atlantic, NAAMES reported no significant differences between the campaign-averaged OMF and sea spray flux during different seasons (Bates et al., 2020), which superficially appears to contradict the underlying assumptions of OCEANFILMS. However, closer examination reveals that NAAMES sampled these variables only at locations and times where OCEANFILMS predicts a minimal influence of marine biology on sea spray. Consequently, the lack of signal observed by NAAMES appears to be fully consistent with OCEANFILMS, which predicts a strong signal would have occurred only during months that NAAMES did not sample. This may help to explain the apparent contradiction between the NAAMES findings and several other laboratory and field studies that have reported changes in sea spray flux in response to marine biology (Fuentes et al., 2010; Alpert et al., 2015; Forestieri et al., 2018; Sellegri et al., 2021). To provide a useful test of seasonal-scale predictions from OCEANFILMS and similar models, observations are needed that constrain sea spray flux and its size-resolved chemistry at a single location, with an adequate sampling of the seasonal cycle. For future field experiments, we recommend that model predictions should be consulted during the planning stages of the experiment, in combination with fundamental and empirical understanding of ocean biogeochemistry processes, to ensure that experiments can provide a useful test of a model's predictions as well as its underlying process-level assumptions.

The most prominent simulated cloud responses to the introduction of MOA emissions are increases in cloud LWP and in-cloud CDNC, especially over the Southern Ocean. These responses lead to a strengthening of cloud shortwave radiative cooling by $-0.36$ W/m$^2$ in the global annual mean, and $-1.6$ W/m$^2$ in the DJF mean over the Southern Ocean. While these responses are consistent with a previous observationally based estimate of MOA cloud impacts McCoy et al. (2015), we note that an implementation of the same parameterization in a different version of E3SM or an Earth system model or global climate model with different representations of aerosol and cloud processes is likely to produce a different cloud response, as has been described for other aerosol sources (e.g., Bodas-Salcedo et al., 2019). Models vary significantly in their sensitivity to cloud–aerosol interactions; the pre-release version of the E3SM used in this study is substantively similar to the CAM5 atmosphere model, which has been shown to exhibit a relatively strong aerosol indirect effect compared with other global climate models of the CMIP5 generation Zelinka et al. (2014).

OCEANFILMS departs from previous parameterizations of sea spray organic emissions in that it is mechanistic rather than empirical. As a result, OCEANFILMS offers a path towards better understanding and representation of the driving mechanisms affecting geographic and seasonal patterns in sea spray organic matter emissions and properties. While the implementation described here uses prescribed ocean biogeochemistry fields (from a prior offline ocean model simulation), future research could explore the possibility of implementing OCEANFILMS or similar approaches with direct, dynamic responses to ocean

biogeochemistry in an Earth system model. A dynamic coupling of ocean biogeochemistry to sea spray aerosol emissions may be required to better understand how MOA emissions may respond to future shifts in ocean biology and chemistry in response to ocean acidification, warming, and changes in circulation patterns.

*Code and data availability.* The implementation of OCEANFILMS described herein is included in the E3SMv1.0 model release, which is publicly archived at https://doi.org/10.11578/E3SM/dc.20180418.36. However, note that while the OCEANFILMS implementation presented here is the same as in E3SMv1, the results presented here were obtained with an earlier version of E3SM, which differs in several important respects from the E3SMv1 release, as described in Section 2.1. The ocean macromolecule distributions used as inputs to OCEANFILMS are archived in the E3SM input data repository at https://web.lcrc.anl.gov/public/e3sm/inputdata/atm/cam/chem/trop_mam/marine_BGC/ monthly_macromolecules_0.1deg_bilinear_latlon_year01_merge_date.nc

*Author contributions.* SMB conceptualized and designed the study. SMB and SME developed the OCEANFILMS parameterization. SMB implemented the OCEANFILMS parameterization in E3SM; RE and XL implemented required changes to the aerosol model infrastructure; additional contributions to the aerosol model development were made by PLM, HW, BS, KZ, and PJR. SMB performed all simulations and analyses. SB prepared the manuscript with contributions from all co-authors.

*Competing interests.* Some authors are members of the editorial board of the journal Atmospheric Chemistry and Physics. The peer-review process was guided by an independent editor, and the authors have also no other competing interests to declare.

*Acknowledgements.* We thank Prof. J. Prospero (U. Miami) and the AEROCOM project for collecting and making available the in situ aerosol measurement network dataset.

Shanlin Wang, Philip Cameron-Smith, and Steve Ghan provided helpful comments on this research and the manuscript.

This research was supported by the Office of Science of the U.S. Department of Energy as part of the Earth System Modeling Program. The Pacific Northwest National Laboratory is operated for DOE by Battelle Memorial Institute under contract DE-AC05-76RL01830. X. Liu acknowledges the funding support of the Office of Science of Department of Energy, Earth System Modeling Program (Award number: DE-SC0011611). This research used high-performance computing resources from the National Energy Research Scientific Computing Center (NERSC), a DOE Office of Science User Facility supported by the Office of Science of the U.S. Department of Energy under Contract No. DE-AC02-05CH11231.

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
