# Peer review of "OCEANFILMS sea spray organic aerosol emissions – Implementation in a global climate model and impacts on clouds"

_Atmospheric Chemistry and Physics, 2018_

## Referee Comment (RC1) · Anonymous Referee #1 · 29 Jun 2018

In this article, Burrows and coauthors describe the implementation of sea-spray organic aerosol emissions into the Energy Exascale Earth Systems Model and the subsequent impact on marine organic aerosol concentrations and cloud properties. While not the first global modeling study to estimate the cloud impacts of marine organic aerosols, this article is one of the most comprehensive to date incorporating an advanced emission parameterization into a state-of-the-science global model. I recommend acceptance of the article to ACP after addressing the minor comments below:

1) After several sensitivity tests, the authors recommend an internally-mixed marine organic aerosol emission sources that is added on top of sea-salt aerosol emissions.

Based on the statistics presented, I don't see how the authors differentiated the performance between the internal and external mixture of sea-salt and organics. While the "ADD" vs. "REPLACE" difference seems more clear, I think that the authors need to do a better job justifying that choice.

2) Another issue that I have with the sensitivity tests is that they seem too dependent on the choice of sea spray emission function. I'd recommend that the author include some evaluation of the sea-salt concentrations; if the predicted sea-salt concentrations agree well with observations at Mace Head and Amsterdam Island, it gives confidence that the sea spray emission function can be used to evaluate the sensitivity tests of the marine organic aerosol emissions.

3) I'd suggest additional description of the mode implementation of the marine organic aerosols. Specifically, I was wondering whether the "REPLACE" and "ADD" sensitivity tests have marine organic aerosols replace/add both number and mass relative to the sea-salt number and mass? A similar question for the "EXT" and "INT" simulation; when you add the marine organics to the primary carbon mode is it really being considered an external mixture in the model. I'm also unclear about the "EXT_REPLACE" simulation, are you adding the same quantities of number and mass to the primary carbon mode as you're removing from the accumulation mode? Does the hygroscopicity of the primary carbon mode change as a function of the amount of marine organic aerosols?

4) Typos: Page 2-Line 27; "ina", Page 5-Lines 7-8; should be "Mårtensson",Page 19-Line 28; "spary", Page 24-Lines 7-11-29; remove bracket from reference, Page 24-Line 18; fix reference, Page 25-Line 20; should be "Nenes, A.,"

5) Figures 3 and 4: consider making them colorblind accessible, especially the red and green lines.

6) Table 5: clarify whether the units are absolute or percentages.

7) Figure 6 and 7: some of the areas have a green color that is not part of the colorscale.

---

## Referee Comment (RC2) · Anonymous Referee #2 · 30 Jun 2018

The manuscript titled "OCEANFILMS sea-spray organic aerosol emissions – Part 1: Implementation and impacts on clouds" by Burrows et al. describes the implementation of the OCEANFILMS parameterization in an earth system model, discusses the results from a set of sensitivity simulations with varying assumptions about the emission state of marine organic aerosol, and examines the impact of such aerosol particles on cloud properties. Two different aspects of the emission state are examined: whether the emitted organic aerosol is internally or externally mixed, and whether it replaces or adds to the amount of sea salt emitted. Internally mixed organic aerosol that adds to the sea salt emission is concluded to agree best with observed seasonal cycles, and particles emitted under such assumptions is found to induce a significant increase in

the in-cloud cloud droplet number concentration, particularly over the southern ocean.

The manuscript is a valuable contribution to the field, as OCEANFILMS is the one of the only physically based parameterizations for organic sea spray aerosol emission, and advancement in its implementation and development is beneficial for subsequent studies. The study also addresses an important open question concerning marine organic aerosol, namely the state in which these particles are emitted, and shows interesting results. Together with its thorough consideration of previous studies on the subject, the manuscript is thus recommended for publication after addressing some comments as suggested below.

Minor comments:

1. Section 3.1.1: is it already conclusive from experimental observations that marine organic matter are internally mixed with sea salt? If so, what's the argument for investigating this aspect in this study?

2. There are quite a lot of nice results, but the interpretations are sometimes lacking. For instance for table 4, the text in the manuscript did not provide any interpretation or connections with the rest of the study beyond stating the values already listed in the table. Figure 8 is another example where perhaps more can be said.

3. Page 13, line 12: any speculations for the different findings in the Southern Ocean and the Arctic? In particular, the correspondence between CCN and IWP changes is not as clear in the Arctic as over the Southern Ocean. Is it still valid to claim that an increase in CCN concentration results in an increase in IWP in the Arctic?

4. Second paragraph in section 4.3.1: does cloud areal fraction include the temporal component? If not, does that change? (cloud lifetime, occurrence frequency)

5. To what degree can the results found in this study be generalized to other models?

6. Related to the previous question, but particularly relevant for conclusions drawn with regards to the impact on clouds: how does E3SM compare to other models and observations in terms of its global aerosol distribution?

Technical comments:

1. The use of acronyms can be more consistent (they are often introduced but not consistently used thereafter). e.g. MOA, Nd, experiment short names (especially in subsequent figures)

2. Figure panel titles can be more useful.

3. One of the main strengths of OCEANFILMS is its physically-based approach. It requires therefore fields from an ocean biogeochemistry model. Despite this, the current study which is advertised as an implementation of OCEANFILMS into an ESM, only utilizes atmosphere-only simulations. This is a reasonable choice for the purpose of performing sensitivity studies, but it raises the question whether OCEANFILMS can indeed be run with current state-of-the-art ESMs or if a more sophisticated ocean biogeochemistry model would be needed. Perhaps it would be good to mention this in the manuscript, also providing information about which ocean biogeochemistry model was used in the current setup (page 5, line 21).

4. Section 3.1: there are 4 sensitivity cases, but only the two mixing state options are explained.

5. Section 4.3: the title states that this section focuses on the INT_ADD simulation, but the content includes discussions of results shown in Figure 3 for the various sensitivity simulations. In general the structure of the results section is not

very consistent, especially with Figure 3 already showing interesting results concerning the impact of marine aerosol on clouds that are worth discussing. The structure of the sections are however divided such that the impact on clouds is only meant to be discussed for the INT_ADD case. Maybe some restructuring of the results section could be beneficial. Or change the contents of the sections to be consistent with the titles.

6. Page 2, line 10-11: "to the total of and..."?

7. Page 3, line 11: "or" –> "of"?

8. Page 7, line 12: MOA density reference?

9. Page 8, line 3: it has been suggested "that" ...

10. Figure 7: significance test in shading?

11. Figure 8: shading in top panels missing?

12. Bibliography errors: spurious "{ }" brackets (page 24 lines 7, 11, 29), author list on line 18 (which is in review from 2016?), title of companion paper also confusing (page 22, line 30: also Part 1?)
* * *

---

## Referee Comment (RC3) · R.H. Moore (Referee) · 4 Jul 2018

**General Comments:**

The manuscript presents changes in simulated organic aerosol, CCN, and cloud microphysical/radiative properties from the OCEANFILMS emissions parameterization incorporated into the E3SM atmosphere model. Ocean properties input to OCEANFILMS are monthly mean simulated values from an unknown biogeochemistry model. A control model run with no marine organics was carried out, and the results from this run are compared to those from 4 different perturbation runs, which vary with regard to how the organic aerosol mass emissions are incorporated into one of four aerosol size

modes. Simulation types include 1) internal vs. external mixing of the organic aerosol and 2) whether the organics add to vs. partially replace sea salt emissions (presumably on a mass basis). A major conclusion of the paper is that the ADD cases are more physically realistic than the REPLACE cases because the former seem to exhibit more pronounced seasonal variability than the latter; such seasonal variation might be expected in some regions from past observational studies (e.g., the Southern Ocean east of Cape Horn, Mace Head). The internal mixing ADD simulation is concluded to be the most realistic and is used for further analysis of simulated perturbations of cloud and radiative forcing properties (relative to the control). The heavy reliance on annual means (despite strong seasonality in modeled organic contributions for the ADD simulations) and on zonal means (despite hemispheric differences in ocean vs. land contributions) may obscure more significant differences between the OCEANFILMS simulation and the control simulation.

The work is interesting and relevant to Atmospheric Chemistry and Physics. The manuscript is well written with a nice introductory review of the literature that contextualizes the present work. However, the current manuscript also 1) lacks important details regarding the simulation experiment setup, 2) fails to adequately support some of its conclusions, and 3) references a companion paper whose relationship to the present work is unclear. I think that this is nice work, and the manuscript should be suitable for publication after the following major/minor comments are adequately addressed:

**Major comments:**

1) Much of the results of this study are presented in terms a percent change in aerosol, CCN, and cloud microphysical/radiative properties from the OCEANFILMS simulations relative to the control simulation (with no marine organic aerosol emissions). Because

only changes are presented, this makes it difficult to assess whether or not the model simulations are producing realistic results and how important a change of, e.g., the 30 CCN cm$^{-3}$ mentioned in the abstract is relative to the background CCN concentration. An in preparation companion paper with some of these results is referenced on Pg. 3, Lines 25-27, but I think it's important to include them here. Please add a column to Figures 5, 6, and 7 showing the INT-ADD simulation result variables to complement the absolute and percent change of these variables. How reasonable are the simulated aerosol number, CCN number, cloud droplet number, and LWP spatial distributions as compared to other model studies or satellite observations? This is important, because ensuring that the model output is realistic is a prerequisite to quantifying the significance of changes from incorporating the new parameterization.

2) The companion paper of Burrows et al., 2017 (referenced on Pg. 3, Lines 25-27 and Pg. 14, Lines 30-32) appears to have a very similar title to the present manuscript and is also a "Part 1". I don't understand the need for a two-part manuscript, and suggest that the authors consider removing these references, removing the Part 1 from the title of the current paper, and incorporating the additional simulation results requested in 1 above into the current paper. If there's a good reason for a two-part paper, then it would be helpful for the authors to provide that rationale in their response and a copy of the other paper so that they can be reviewed together. Often, a stand alone paper followed by a later, independent paper is a better path than a two-part set of manuscripts.

3) The color scales of the figures are very large and make it difficult to discern small changes in the statistically-significant ocean regions, while emphasizing large changes over the continents. Please mask the continental portions of the maps and change the color scale extents so that the regions of interest are more discernable. Figure 6-top-right is a good example of this, where it is hard to tell if LWP varies in the one meaningful shaded region east of Cape Horn by closer to 0% or closer to 60%. Similarly, the accumulation mode number hotspot in east Africa in Fig. 5 dominates that scale at the expense of the North Atlantic and Southern Oceans (where aerosol changes appear

to be statistically-significant but close to 0 kg$^{-1}$).

4) Does the model provide CCN information at higher supersaturations than 0.1%, and how do the organic emissions from OCEANFILMS change the shape of the CCN spectrum (i.e., CCN concentration or activated fraction as a function of supersaturation)? How much variability in modeled supersaturation is observed in the model results? Is 0.1% representative of remote regions, or are larger supersaturations found where there are low aerosol concentrations?

5) In Section 2.3, it is stated that the sea spray emissions parameterizations of Martensson et al. (2003) and Monahan (1986) were based on lab experiments that included only inorganic species and no organic species, so it is assumed that these parameterizations do not include any contributions from organic matter. This line of reasoning doesn't make sense to me, as presumably the presence of dissolved organic matter in the seawater would contribute sea spray aerosol emissions from the same bubble bursting processes that produce the inorganic aerosol, but not necessarily in the same amount or size. Monahan's parameterization is limited to particles above 2.5 $\mu$m, where perhaps neglecting an organic mass contribution is reasonable, but Martensson et al.'s parameterization extends down to 20 nm, where the likelihood of this is assumption being true appears less convincing. On what basis can it be postulated that the emission of organic and inorganic species from the same mechanistic process (i.e., bubble bursting) can be treated independently of each other?

6) In Section 2.4, some of the key inputs from the biogeochemical model to the OCEANFILMS parameterization are mentioned briefly. While the detailed model discussion is given by Burrows et al. (2014), it would also be helpful to have a few more details given here. For example, which biogeochemistry model was used to compute the monthly mean input fields? What parameters from the E3SM atmosphere model, if any, serve as inputs to the combined OCEANFILMS and sea spray emissions parameterizations (e.g., wind speed)? If OCEANFILMS determines the OMF of the emitted sea spray aerosol, but the sea spray aerosol emissions rate is governed by parameterizations based on no organic emissions, how are these parameterizations blended together in the ADD vs. REMOVE simulations? Is this blending done on a mass or number basis (or even at all - did I misunderstand something here)?

7) Section 3.1 currently lacks details on how the ADD versus REMOVE simulations are set up and the meaning of what these simulations represent. For example, am I to understand that if 1 $\mu$g of MOA is added to the accumulation mode then 1 $\mu$g of NCL is subtracted from the accumulation mode in the REMOVE simulations? Under what circumstances might that make physical sense?

8) There appear to be large differences in the organic aerosol emissions associated with the ADD vs. the REMOVE simulations (Table 4). Shouldn't at least the sum of the MOA mass emissions rates across all size modes (in Gg/yr) be the same in all 4 simulations? I could see how differences in how these mass emissions are partitioned across mixing state, size, number might drive differences in mean atmospheric burdens (Figure 2 and Table 4), but I don't understand what would change the mass emissions rates themselves. Put another way - if the ocean biogeochemistry model monthly means drive OCEANFILMS, and OCEANFILMS drives the emissions rates, why would these emissions rates vary across the four simulations that incorporate the OCEANFILMS parameterizations?

9) Sections 3.1.1 and 3.1.2 seem out of place here as they provide background from previous observational studies. I think that this discussion would be better suited in the "Introduction" section rather than the "Model simulations and analysis methods" section.

10) The conclusion drawn on Pg. 15, Lines 4-5 seems very speculative based on the results presented in this manuscript. Despite the good agreement at Mace Head, the comparisons with observational data from Amsterdam Island and Point Reyes don't appear very conclusive. Given the speculativeness of the writing, I suggest that the authors strike this paragraph, and proceed with Sections 4.2 and 4.3 as an example

case for closer examination. It might also be worth noting that while the ADD-INT case is presented, the ADD-EXT case would likely yield similar results.

11) I think that the conclusions drawn on Pg. 1, Line 7 and Pg. 20, Line 3 about "best agreement" or "most realistic" are too strong and not supported by the findings of this study. Instead, I suggest that these statements be reworded to note that: the ADD organics simulations show seasonal variation in organic aerosol that is consistent with some past observational studies, while the REMOVE organics simulations lack this pronounced seasonality. The assumption of an internally or externally mixed organic aerosol appears to have a minor, secondary effect on the simulation results (Figure 2, Table 4).

12) Since the continents have a greater contribution to the zonal mean in the northern hemisphere than in the southern hemisphere, would recomputing Figure 8 with the continental grid boxes excluded reveal a stronger (larger) period of statistical significance in the Southern Ocean and some period of significance in the North Atlantic and Pacific Oceans?

**Minor comments:**

Pg. 2, Line 11: "of and" missing words

Pg. 2, Line 27: "ina" typo

Pg. 3, Line 26: If this paper is still in preparation, please cite it as such rather than with a publication year.

Pg. 5, Line 29: While "renaming" is the model process of making the jump from one size mode to another, it isn't really a physical process. I suggest removing it from this parenthetical list, but retaining the good, later discussion on Pg. 6, Line 13-14.

Pg. 14, Line 9: Please add a citation for the assumption that OM:OC ∼ 1.9.

Pg. 19, Line 28: "sea-spary" typo

Pg. 20, Line 29: May wish to update code availability statement if source code was released earlier this year.

Consider reversing Tables 1 and 2, as the abbreviations that are first defined in Table 2 are used in Table 1.

Table 1: Why does the density of MOA have four significant figures, while the other species have at most 2?

Figure 8: The blue (or black) trace in the top row of panels is very difficult to discern from the green trace. Suggest making it dashed and in front?

[Figure]

---

## Author Comment (AC1) · 6 Dec 2021

We thank the reviewers for their time and efforts in providing constructive comments and feedback on our manuscript.

Overall, the comments made it clear to us that the original manuscript did not adequately communicate that the version and configuration of the E3SMv0 atmosphere model used here is nearly identical to variants of CAM5.3 that have previously been evaluated elsewhere in the peer-reviewed literature, and included in model intercomparisons.  Additionally, due to the large amount of material, we had planned to include some more specific evaluations in a "Part 2" manuscript, but this led to gaps in the information available to readers in the present manuscript.  We have therefore made two sets of changes to the manuscript to address these comments:

1. We have added some text to the introduction that clarifies the relationship between the E3SMv0 Atmosphere Model and CAM5.3, briefly summarizes the results of the evaluations of this model's physics, and directs readers to additional publications documenting those evaluations and comparisons with other global models.  In particular, the reviewers have requested more information about the behavior of the clouds and aerosols in general in this model. The new text more clearly explains that the model's baseline aerosol and clouds had already been documented in previous publications (which we point readers to), and that we therefore do not repeat that evaluation here.

2. We have introduced a new section, "Model evaluation with observational data", which includes additional model evaluations that are more specific to the topic of this manuscript.  In doing this, we have also removed dependencies on the originally planned "Part 2" paper, and following the suggestion of referee Richard Moore, the paper title has also been revised to indicate that this is now a stand-alone contribution.

   Specifically, the revised manuscript includes comparisons to observations of aerosol mass for sea salt, dust, and total aerosol, including seasonal cycles of sea salt. Additionally, it includes a comparison with observed TOC at marine sites, including seasonal cycles where available.

For clarity, it is perhaps worth noting that the atmosphere model in E3SMv1, by contrast, does differ in several significant respects from the model used here (E3SMv0) and from CAM5.3.  In particular, E3SMv1 uses the CLUBB parameterization of boundary-layer turbulence and convection, and has shifted from 30 to 72 vertical layers; these changes have significant impacts on the clouds and atmospheric physics in E3SMv1, and will be documented in future publications.  As a consequence, the findings of this manuscript therefore do not necessarily apply exactly within the E3SMv1 model.  We expect any differences in the E3SMv1 model representations of aerosol-cloud interactions and convection in general to be responsive to this new aerosol source, but the focus of this paper is the implementation of marine organic sea spray aerosol and the model's responsiveness to it within the E3SMv0 atmosphere model.

Finally, following up on the referee comments, we reorganized the manuscript and revised the section headings to improve clarity and structure, and we have revised the title to reflect that this is now a standalone paper.

Responses to specific reviewer comments follow.

**Anonymous reviewer 1:**

Based on the statistics presented, I don't see how the authors differentiated the performance between the internal and external mixture of sea-salt and organics. While the "ADD" vs. "REPLACE" difference seems more clear, I think that the authors need to do a better job justifying that choice.

We have revised and reorganized the description of the sensitivity cases to more clearly explain how the internal versus external mixing states are accomplished.

2) Another issue that I have with the sensitivity tests is that they seem too dependent on the choice of sea spray emission function. I'd recommend that the author include some evaluation of the sea-salt concentrations; if the predicted sea-salt concentrations agree well with observations at Mace Head and Amsterdam Island, it gives confidence that the sea spray emission function can be used to evaluate the sensitivity tests of the marine organic aerosol emissions.

We agree with the reviewer that including some evaluation of simulated sea salt in the manuscript is useful. The new section 4, "Model evaluation with observational data", includes an evaluation of the predicted sea salt concentrations, which we had originally planned to include in a second, companion paper. For this evaluation, we compare simulated sea salt against the AEROCE/SEAREX dataset, a dataset widely used for evaluation and benchmarking of sea spray aerosol concentrations in global models. The newly introduced Figure 2, Figure 3, and Figure 4 show a map of the locations of the sampling sites (Figure 2), a scatterplot of the model versus observations for climatological monthly mean Na mass, seasonal cycles at the sites. The model systematically overpredicts sea spray amount relative to this dataset, but is typically within an order of magnitude, which is a reasonable level of agreement to expect for this type of comparison (comparison of a global model versus local observations of sea spray aerosol mass). E.g, see results from Tsigardis et al. (2013) reproduced below.

Tsigaridis, K., Koch, D. and Menon, S., 2013. Uncertainties and importance of sea spray composition on aerosol direct and indirect effects. *Journal of Geophysical Research: Atmospheres*, *118*(1), pp.220-235.

[Figure]

3) I'd suggest additional description of the mode implementation of the marine organic aerosols. Specifically, I was wondering whether the "REPLACE" and "ADD" sensitivity tests have marine organic aerosols replace/add both number and mass relative to the sea-salt number and mass? A similar question for the "EXT" and "INT" simulation; when you add the marine organics to the primary carbon mode is it really being considered an external mixture in the model. I'm also unclear about the "EXT_REPLACE" simulation, are you adding the same quantities of number and mass to the primary carbon mode as you're removing from the accumulation mode? Does the hygroscopicity of the primary carbon mode change as a function of the amount of marine organic aerosols?

We thank the reviewer for pointing out that some additional clarification would be helpful here. Please see our response to the similar remark above (#1) and associated changes in the text regarding the distinction between EXT and INT.

We also add the following text to explain the ADD and REPLACE assumptions:

"Regarding the assumption of replacing versus adding sea salt, we again make two extreme choices. In REPLACE cases, the mass of emitted sea salt is reduced by an amount that is equal to the emitted MOA mass, such that the total emitted aerosol mass remains constant. The number of emitted sea salt particles is also reduced proportionally in each mode. In contrast, in the ADD cases, the mass and number of emitted sea salt are unchanged from the BASE model. Emitted MOA mass is added into the respective aerosol modes, increasing both the total mass and the total number of emissions in that mode. Note that either the addition of MOA mass, or the replacement of sea salt by MOA, will impact the volume-weighted hygroscopicity of that mode, which is used in the droplet activation scheme (see Section 2.6 and Table 2)."

4) Typos: Page 2-Line 27; "ina", Page 5-Lines 7-8; should be "Mårtensson",Page 19- Line 28; "spary", Page 24-Lines 7-11-29; remove bracket from reference, Page 24-Line 18; fix reference, Page 25-Line 20; should be "Nenes, A.,"

Thank you for catching these typos; we have corrected them.

5) Figures 3 and 4: consider making them colorblind accessible, especially the red and green lines.

Since the lines in these plots are distinguished not only by color, but also by line pattern and thickness, they should already be accessible to readers with common forms of color blindness.

6) Table 5: clarify whether the units are absolute or percentages.

To make this clearer, in the revised manuscript we state in the table caption that these are relative (percentage) changes – in addition, this was already stated in the table heading, and we still retain this, so readers can now find this information in both locations.

7) Figure 6 and 7: some of the areas have a green color that is not part of the colorscale.

Thanks for pointing this out. We have fixed the contours in these figures to be more legible.

**Anonymous reviewer 2:**

The manuscript titled "OCEANFILMS sea-spray organic aerosol emissions – Part 1: Implementation and impacts on clouds" by Burrows et al. describes the implementation of the OCEANFILMS parameterization in an earth system model, discusses the results from a set of sensitivity simulations with varying assumptions about the emission state of marine organic aerosol, and examines the impact of such aerosol particles on cloud properties. Two different aspects of the emission state are examined: whether the emitted organic aerosol is internally or externally mixed, and whether it replaces or adds to the amount of sea salt emitted. Internally mixed organic aerosol that adds to the sea salt emission is concluded to agree best with observed seasonal cycles, and particles emitted under such assumptions is found to induce a significant increase in C1 ACPD Interactive comment Printer-friendly version Discussion paper the in-cloud cloud droplet number concentration, particularly over the southern ocean. The manuscript is a valuable contribution to the field, as OCEANFILMS is the one of the only physically based parameterizations for organic sea spray aerosol emission, and advancement in its implementation and development is beneficial for subsequent studies. The study also addresses an important open question concerning marine organic aerosol, namely the state in which these particles are emitted, and shows interesting results. Together with its thorough consideration of previous studies on the subject, the manuscript is thus recommended for publication after addressing some comments as suggested below.

Minor comments:

1. Section 3.1.1: is it already conclusive from experimental observations that marine organic matter are internally mixed with sea salt? If so, what's the argument for investigating this aspect in this study?

We thank the reviewer for this comment, which showed us that the original manuscript text was not clear on this point. The results from experimental studies summarized by Prather et al. (2013) have shown that there are four distinct particle types in sea spray aerosol: pure sea salt particles, mixed sea salt – organic carbon particles, biological particles, and pure organic carbon particles. Of these four types, the pure organic carbon type dominate for particles < 80 nm diameter, which corresponds to the Aitken mode in MAM4. Between 80 nm and 1 um, which is the accumulation mode size range in MAM4, the internally-mixed sea salt – organic carbon type dominates at most particle sizes, indicating internal mixture. However, a significant fraction of pure organic carbon particles are still present at the smaller end of this size range, and at the upper end of this size range (around 1 um), about 20% of particles were characterized as pure salt. Among particles larger than about 1.5 um, most were pure sea salt (which is consistent with our choice to not emit MOA in the coarse mode in this implementation of OCEANFILMS).

Given this experimental evidence, it is clear that sea spray particles emitted in the MAM4 accumulation mode size range contain both internally mixed and externally mixed particles. Since the experiments indicate that majority of these particles are likely internally mixed, the internal mixing assumption appears to be most consistent with experimental evidence. However, given that some externally-mixed particles are also present in this size range, treating all particles

as internally mixed is a simplification, and it is useful to test the model's sensitivity to this simplifying assumption.

We have modified the text as follows:
"Experiments in which sea spray aerosol is generated by breaking waves in the presence of induced phytoplankton blooms provide the most realistic physical model of the sea spray aerosol production process (Ault et al., 2013; Collins et al., 2013; Prather et al., 2013). These experiments, combined with single-particle mass spectrometry and electron microscopy, have shown that the smallest emitted particles (up to about 100~nm in diameter) are primarily organic, and marine organic matter is typically mixed internally with sea salt upon emission for intermediate sizes (from about 200 nm to 1 um in diameter). The vast majority of the largest particles (greater than 1.5 um in diameter) are composed almost entirely of inorganic salts, with inorganic salts comprising about 20% of emitted particles with diameters close to 1 um. Since the MAM4 accumulation mode extends from 80 nm to 1 um, the majority of particles emitted in this size range are likely internally mixed, but some externally-mixed particles (composed either of purely organic materials or purely inorganic salts) are also emitted in this size range. Therefore, while the internal mixing assumption is likely more consistent with current experimental evidence, it is also important to understand the impact of this simplifying assumption through the mixing state sensitivity cases."

2. There are quite a lot of nice results, but the interpretations are sometimes lacking. For instance for table 4, the text in the manuscript did not provide any interpretation or connections with the rest of the study beyond stating the values already listed in the table. Figure 8 is another example where perhaps more can be said.

We thank the reviewer for this suggestion; we agree that it would be useful to include some additional interpretation of these particular results in the text.

For Table 4, we have revised the text as follows (added text in bold face):

"Table 4 shows the global atmospheric residence times, with respect to all removal processes, for marine organic aerosol in the Aitken, accumulation and primary carbon modes, in each sensitivity case. The mean global residence time ranges from 0.29 – 0.33 days for the Aitken mode, and 1.1 – 1.2 days for the accumulation and primary carbon modes, with small variations across sensitivity cases. **In the Aitken mode, losses to dry deposition predominate, while losses in the accumulation mode are primarily due to wet deposition. These lifetimes are shorter than those computed for continental emissions of aerosols such as dust in similar models (Wu et al., 2020), which is consistent with expectations for marine-emitted aerosols (Burrows et al., 2013). Most global climate models produce frequent light rain in marine regions, which strongly controls the rate of aerosol wet removal over oceans, and which may produce biases in the geographic distribution of simulated aerosol burden, especially over oceans (Wang et al., 2021). Recent work by Emerson et al. (2020) also suggests that the dry deposition parameterization used in E3SM may overestimate the rate of dry deposition to water surfaces in the Aitken and accumulation modes. However, since the simulated effects on CCN and clouds are controlled mostly by the accumulation mode, model biases in dry deposition have only minor effects on the main results of this study."**

For the original Figure 8 (note figure numbering has been revised), we have revised the text as follows, to more clearly explain the reasoning behind these significance tests, and their interpretation:

"To place these findings in context, it is important to note that Figure 9 and Figure 11 attempt to evaluate the statistical significance of a comparatively small signal in the presence of significant noise at the grid-box level. It is a well-established and characteristic behavior of free-running atmospheric model simulations that local perturbations will rapidly produce large differences in the model's physical state due to chaotic dynamical responses, producing noisy model responses (Lorez et al., 1963; Wan et al., 2017). This noise can obscure the signal of simulated impacts in the model, unless that signal is locally very strong. One strategy for overcoming this signal-to-noise issue is to perform averaging over larger regions, e.g., to determine whether a robust signal can be detected in the zonal or average mean. This is analogous to the familiar "signal averaging" approach used in many experimental fields, which takes advantage of the fact that when averaging across many samples (in this case, model grid boxes), a physical signal will tend to accumulate, while random noise tends to be reduced by the averaging process (Hassan and Anwar, 2010).

We therefore explore the cloud impacts further for the INT_ADD case by examining the changes in zonal mean cloud cover for statistical significance. As shown in Figure 12, we find that over the Southern Ocean, despite the model's internal variability, the simulation produces robust zonal mean changes in SWCF over the Southern Ocean that are significant at the p<0.5% level during all seasons, and at the p<0.1% level during austral summer (DJF) and autumn (MAM). This is a key finding of the present study, and is consistent with the top-down constraints on changes in zonal mean SWCF from (McCoy et al., 2015), lending increased confidence that both the model result and the values inferred from the satellite observations are realistic."

3. Page 13, line 12: any speculations for the different findings in the Southern Ocean and the Arctic? In particular, the correspondence between CCN and IWP changes is not as clear in the Arctic as over the Southern Ocean. Is it still valid to claim that an increase in CCN concentration results in an increase in IWP in the Arctic?

The paragraph the reviewer is referencing here discusses the model's liquid water path (LWP) response to changes in CCN number in the different sensitivity cases. (The reviewer wrote IWP, but since we did not show or discuss ice water path results, we assume they meant LWP).

The reviewer has correctly pointed out that the signals/responses are much weaker in the Arctic, but there is an increase in LWP resulting from the increase in summertime CCN (in the EXT_ADD case). The weaker Arctic cloud responses (relative to the Southern Ocean) likely result in large part from the weaker increase in CCN number in the Arctic (as shown in the original Figure 3, top row), which is in turn due to the larger emissions of sea spray organic matter in the Southern Ocean, as compared to the Arctic (original Figure 2). The comparison between the Arctic and Southern Ocean is also complicated by the fact that the Arctic region contains significant areal fractions of land mass, where we do not expect to see large responses from changes to sea spray emissions.

Also, although the model does show increases in the summertime Arctic LWP in response to CCN changes, our significance testing on the LWP variable (not shown in the manuscript) indicated that the changes in zonal mean LWP (and cloud droplet number concentration) were indistinguishable from natural variability in the Arctic – in contrast to the Southern Ocean, where statistically significant changes occurred in both variables. This is also, in large part, why we focus less attention on the Arctic region in the subsequent discussion. We have added the following text to more clearly indicate this to readers:

"In particular, the zonal mean values of CDNC and LWP both increase over the summertime Southern Ocean in the ADD and EXT cases. Zonal-mean LWP also increases in the summertime Arctic in the EXT_ADD case, but decreases in the REPLACE cases. The changes in these two variables were clearly distinguishable from natural variability over the Southern Ocean (p<0.1%), while they were largely not distinguishable from natural variability over the Arctic (t-test calculations for significance not shown). The stronger cloud responses over the Southern Ocean (as compared to the Arctic) are likely attributable largely to two related causes. First, the presence of significant landmass in the Northern Hemisphere means that the oceanic area where changes in sea spray emissions can take effect is significantly smaller. Second, the emissions of sea spray organic matter are smaller in the Arctic relative to the Southern Ocean (Figure 8), resulting in smaller changes in zonal mean CCN (Figure 9, top row). Finally, it is possible that cloud sensitivity to changes in CCN concentrations differs between the Arctic and the Southern Ocean."

Also, in revising this section to address the reviewer's comment, we noticed that the original manuscript did not show our analysis of seasonal changes in the model's simulated zonal mean CCN and cloud droplet number concentration (CDNC). The changes in zonal mean CCN (S=0.1%) are statistically significant in the summertime Arctic, but are much smaller in magnitude than the changes in CCN over the Southern Ocean:

[Figure]

This difference is likely is the primary reason that cloud responses are also weaker in the Arctic. In the Arctic, a summertime increase in CDNC is observed, but it is debatable whether these changes are distinguishable from natural variability. In contrast, in the Southern Ocean, the simulated changes are large and statistically significant during austral spring, summer and

autumn, as shown in this figure (which is analogous to Figure 8 of the original manuscript, but for zonal mean cloud droplet number concentration):

[Figure]

In the subpolar region from approximately 40-60N, some changes in zonal mean cloud droplet number concentration appear to be significant at the p<5% level (lowest blue line) and at the p<1% level (central blue line). However, from 60-90N the changes largely appear to not be statistically significant.

Changes in zonal mean cloud LWP are small relative to the absolute values, but also pass our threshold for statistical significance over the Southern Ocean (but not the Arctic):

[Figure]

We have added these figures in the revised manuscript, as additional panels, since they help to elucidate the causes of the changes in zonal mean short-wave cloud forcing.

4. Second paragraph in section 4.3.1: does cloud areal fraction include the temporal component? If not, does that change? (cloud lifetime, occurrence frequency)

As the reviewer's comment points out, changes in cloud areal fraction -- a value that is easily obtainable from our model -- can arise from changes in either the extent, lifetime, or occurrence frequency of individual clouds – values that are not so readily obtainable from our model or

other similar models. Developing a methodology to diagnose these different effects within the model (e.g., by tracking individual clouds) is beyond the scope of this paper.

Following up on this comment, we have added a sentence to the text to more explicitly highlight this point for readers:
"Note that changes in these annual-mean cloud variables can be caused by either changes in the properties of individual clouds, or changes in the temporal behavior of clouds, i.e., frequency of cloud formation and occurrence, and cloud lifetime, or a combination of both."

5. To what degree can the results found in this study be generalized to other models?

This is certainly a fair question, but difficult to answer, since representations of aerosols, clouds, and aerosol-cloud interactions can vary significantly between models. Therefore, it is likely that if the OCEANFILMS parameterization were implemented in other climate or Earth System models, those models would exhibit a different cloud response. For example, Bodas-Salcedo et al. (2019) found that an older version of the HadGEM3 model showed a significant climate response to changes in wind-induced emissions of dimethyl sulfide (and the resulting changes in marine cloud condensation nuclei). However, a newer version of the same model, with modifications to the aerosol and cloud microphysics schemes, did not show similar impacts.

Answering this question would likely require a broader community effort that included comparisons between the marine aerosol treatments in different models, and their climate impacts as simulated by those models. This question may be answered by future studies if/when additional models implement the OCEANFILMS parameterization, and evaluate its impacts on clouds and/or compare it with other parameterizations of marine organic matter within those models.

It is also for this reason that we use the observationally based analysis of McCoy et al. (2015) as a point of references for the global radiative impacts of the marine organic aerosol. McCoy et al. (2015) inferred that that increases in cloud albedo due to marine organic aerosol contributed approximately 1-2 $W/m^2$ additional annual mean radiative cooling over the Southern Ocean, with the cooling occurring primarily during the austral summer months (DJF). The cloud responses we simulate here are within this range.

Following up on the reviewer's comment, we have added the following sentences to the Conclusions section, which provide more context on how the sensitivity of models to changes in aerosol varies across other similar model (i.e., GCMs).

"While these responses are consistent with a previous observationally-based estimate of MOA cloud impacts (McCoy et al., 2015), we note that an implementation of the same parameterization in a different version of E3SM or an Earth system model or global climate model with different representations of aerosol and cloud processes is likely to produce a different cloud response, as has been described for other aerosol sources (Bodas-Salcedo et al., 2019). Models vary significantly in their sensitivity to cloud-aerosol interactions; the pre-release version of the E3SM used herein is substantively similar to the CAM5 atmosphere model, which

has been shown to exhibit a relatively strong aerosol indirect effect compared with other global climate models of the CMIP5 generation (Zelinka et al., 2014)."

Bodas-Salcedo, A., Mulcahy, J.P., Andrews, T., Williams, K.D., Ringer, M.A., Field, P.R. and Elsaesser, G.S., 2019. Strong dependence of atmospheric feedbacks on mixed-phase microphysics and aerosol-cloud interactions in HadGEM3. *Journal of Advances in Modeling Earth Systems*, *11*(6), pp.1735-1758.

Zelinka, M. D., Andrews, T., Forster, P. M., and Taylor, K. E. (2014), Quantifying components of aerosol-cloud-radiation interactions in climate models, *J. Geophys. Res. Atmos.*, 119, 7599– 7615, doi:10.1002/2014JD021710.

6. Related to the previous question, but particularly relevant for conclusions drawn with regards to the impact on clouds: how does E3SM compare to other models and observations in terms of its global aerosol distribution?

We thank the reviewer for raising this issue.  Thanks to this comment, we realized that the relationship between the model version we used and versions that have been previously documented in the literature was not very clearly explained in the original text.  In brief, the atmosphere model used here is nearly identical to CAM5.3, and the aerosol and cloud representations in that model, as well as the minor modifications we use (other than MOA), have been documented elsewhere.  Therefore we do not repeat the exercise of performing general evaluation of the model's aerosol, clouds, and climate in this study.

In the revised manuscript, we have expanded the model description section to more clearly explain the relationship of the model version used here to CAM5.3 (and to the atmosphere component of the E3SMv1 model, which was released after this study was originally completed).  We have also included additional references to help readers find further information on the performance and behavior of the aerosols, clouds, and climate in the model.

Technical comments:
1. The use of acronyms can be more consistent (they are often introduced but not consistently used thereafter). e.g. MOA, Nd, experiment short names (especially in subsequent figures)

We have had a professional copyeditor work on the manuscript, work that fixing the acronym usage and the formatting of references.

2. Figure panel titles can be more useful.

Thanks for pointing this out.  We have revised the figure panel titles to be more descriptive in several of the figures.

3. One of the main strengths of OCEANFILMS is its physically-based approach. It requires therefore fields from an ocean biogeochemistry model. Despite this, the current study which is advertised as an implementation of OCEANFILMS into an ESM, only utilizes atmosphere-only

simulations. This is a reasonable choice for the purpose of performing sensitivity studies, but it raises the question whether OCEANFILMS can indeed be run with current state-of-the-art ESMs or if a more sophisticated ocean biogeochemistry model would be needed. Perhaps it would be good to mention this in the manuscript, also providing information about which ocean biogeochemistry model was used in the current setup (page 5, line 21).

Thank you for this comment; we agree with this assessment of the reviewer that further work would be required to determine whether this parameterization can indeed be used successfully with a dynamic dependence on simulated ocean biogeochemistry.

In the model description, we have added some details about the ocean biogeochemistry model that was used to generate the input fields for the parameterization, and to more clearly state that these fields are saved in files that are used as prescribed inputs to the model.

"The OCEANFILMS parameterization, introduced and described in detail in Burrows et al. (2014), proposes a mechanistic approach for connecting emissions of sea-spray organic aerosol to models of ocean biogeochemistry. As described previously in Burrows et al. (2014), we used the Parallel Ocean Program (POP; Maltrud et al., 1998) to simulate the ocean's general circulation and its biogeochemical elemental cycling (BEC) routines (Moore et al., 2004) to simulate marine biogeochemistry. Both are components of the Community Earth System Model (CESM; http://www2.cesm.ucar.edu). Calculations of the ocean biogeochemistry fields were performed using the CESM 1.0 beta release 11. Monthly-mean concentrations of five broad classes of macromolecules in ocean surface waters are derived from the POP-simulated distributions of phytoplankton, zooplankton, and semi-labile dissolved organic carbon. The files containing these macromolecular distributions are publicly available as part of the E3SM input data repository (see data availability statement). Chemical and physical properties are assigned to each of these macromolecular classes, based on representative proxy molecules for which laboratory measurements are available. Using a Langmuir isotherm-based approach, OCEANFILMS then predicts the surface coverage of ocean bubble films with each of these model macromolecules. This surface film coverage, together with a prescribed bubble film thickness, determines the OMF of the emitted sea spray aerosol, which is calculated online within E3SM on the basis of the prescribed macromolecule distributions."

We have also revised the "Conclusions and Summary" section to more explicitly point out that we used prescribed input fields here, and that certain research questions would require a dynamic link to ocean biogeochemistry.

"While the implementation described here uses prescribed ocean biogeochemistry fields (from a prior offline ocean model simulation), future research could explore the possibility of implementing OCEANFILMS or similar approaches with direct, dynamic responses to ocean biogeochemistry in an Earth System Model. A dynamic coupling of ocean biogeochemistry to sea spray aerosol emissions may be required to better understand how sea-spray organic aerosol emissions may respond to future shifts in ocean biology and chemistry in response to ocean acidification, warming, and changes in circulation patterns."

4. Section 3.1: there are 4 sensitivity cases, but only the two mixing state options are explained.

Thanks for pointing this out.  In response to this comment and similar comments from the other two reviewers, we have added an explanation of the ADD and REPLACE options.

5. Section 4.3: the title states that this section focuses on the INT_ADD simulation, but the content includes discussions of results shown in Figure 3 for the various sensitivity simulations. In general the structure of the results section is not very consistent, especially with Figure 3 already showing interesting results concerning the impact of marine aerosol on clouds that are worth discussing. The structure of the sections are however divided such that the impact on clouds is only meant to be discussed for the INT_ADD case. Maybe some restructuring of the results section could be beneficial. Or change the contents of the sections to be consistent with the titles.

Thank you for this comment.  We have restructured the results section, and we believe we have fixed the issue with consistency between the titles of sections and their content in the revised manuscript.

6. Page 2, line 10-11: "to the total of and..."?

Thanks, we have fixed this.

7. Page 3, line 11: "or" –> "of"?

Thanks, we have fixed this.

8. Page 7, line 12: MOA density reference?

Thanks for this comment; in the revised manuscript we add text here to explain that we have used the density of alginic acid, a polysaccharide contained in algal cell walls, as a proxy for the density of MOA.

9. Page 8, line 3: it has been suggested "that" ...

Thanks, we have fixed this.

10. Figure 7: significance test in shading?

Thanks for pointing out that this was not clear in the figure captions.  We have updated the caption of the figure that was originally numbered as Figure 6 to clarify that the shading indicates significance testing.  The original Figure 7 has been removed in the revised manuscript.

11. Figure 8: shading in top panels missing?

The shading is present in the upper panels of original Figure 8.  It is visually not very prominent because the regions of statistically significant differences in zonal mean SWCF are very limited, and the differences are small relative to the absolute values of the zonal mean SWCF.  In the

revised manuscript, we have added additional panels to this figure, showing the zonal mean changes in CCN (S=0.1%), cloud drop number concentration, and cloud liquid water path.  Since the relative differences are more prominent for CCN and cloud droplet number concentration, their inclusion should help to make the presence of shading, and the interpretation of the figure more apparent.

12. Bibliography errors: spurious "{ }" brackets (page 24 lines 7, 11, 29), author list on line 18 (which is in review from 2016?), title of companion paper also confusing (page 22, line 30: also Part 1?)
Thanks; we have fixed these issues.

**Richard Moore (referee):**

General Comments: The manuscript presents changes in simulated organic aerosol, CCN, and cloud microphysical/radiative properties from the OCEANFILMS emissions parameterization incorporated into the E3SM atmosphere model. Ocean properties input to OCEANFILMS are monthly mean simulated values from an unknown biogeochemistry model. A control model run with no marine organics was carried out, and the results from this run are compared to those from 4 different perturbation runs, which vary with regard to how the organic aerosol mass emissions are incorporated into one of four aerosol size C1 modes. Simulation types include 1) internal vs. external mixing of the organic aerosol and 2) whether the organics add to vs. partially replace sea salt emissions (presumably on a mass basis). A major conclusion of the paper is that the ADD cases are more physically realistic than the REPLACE cases because the former seem to exhibit more pronounced seasonal variability than the latter; such seasonal variation might be expected in some regions from past observational studies (e.g., the Southern Ocean east of Cape Horn, Mace Head). The internal mixing ADD simulation is concluded to be the most realistic and is used for further analysis of simulated perturbations of cloud and radiative forcing properties (relative to the control). The heavy reliance on annual means (despite strong seasonality in modeled organic contributions for the ADD simulations) and on zonal means (despite hemispheric differences in ocean vs. land contributions) may obscure more significant differences between the OCEANFILMS simulation and the control simulation.

The work is interesting and relevant to Atmospheric Chemistry and Physics. The manuscript is well written with a nice introductory review of the literature that contextualizes the present work. However, the current manuscript also 1) lacks important details regarding the simulation experiment setup, 2) fails to adequately support some of its conclusions, and 3) references a companion paper whose relationship to the present work is unclear. I think that this is nice work, and the manuscript should be suitable for publication after the following major/minor comments are adequately addressed:

Major comments:

1) Much of the results of this study are presented in terms a percent change in aerosol, CCN, and cloud microphysical/radiative properties from the OCEANFILMS simulations relative to the control simulation (with no marine organic aerosol emissions). Because only changes are presented, this makes it difficult to assess whether or not the model simulations are producing realistic results and how important a change of, e.g., the 30 CCN cm−3 mentioned in the abstract is relative to the background CCN concentration. An in preparation companion paper with some of these results is referenced on Pg. 3, Lines 25-27, but I think it's important to include them here. Please add a column to Figures 5, 6, and 7 showing the INT-ADD simulation result variables to complement the absolute and percent change of these variables. How reasonable are the simulated aerosol number, CCN number, cloud droplet number, and LWP spatial distributions as compared to other model studies or satellite observations? This is important, because ensuring that the model output is realistic is a prerequisite to quantifying the significance of changes from incorporating the new parameterization.

We thank the reviewer for this thoughtful comment. The comment contains several points, which we address individually.

Regarding the companion paper: Following the recommendations of the referees, the revised manuscript includes more of the model evaluation results that we originally planned to include in a companion manuscript (the companion manuscript is no longer planned).

Regarding the general behavior of the aerosol and clouds in the model, please see our comment in response to a similar comment from Reviewer 2. In brief, these general features of the model had been documented in prior publications, and therefore we do not find it necessary to repeat that documentation in this publication, which is focused on the implementation of the new OCEANFILMS parameterization and the model's cloud responses to it. However, the comments of the reviewers alerted us that this was not clearly communicated in the original manuscript text. In the revised manuscript, we have expanded the model description to more clearly identify the model's relationship to CAM5.3 (almost identical in this pre-release version of E3SM), and to more clearly point out the previous publications that document the overall features and behavior of the model's aerosols, clouds, and climate.

We have also added the requested column showing the simulation results from the CNTL (rather than the INT_ADD) simulation, which complements the contour plots of the relative and absolute changes between CNTL and INT_ADD.

2) The companion paper of Burrows et al., 2017 (referenced on Pg. 3, Lines 25-27 and Pg. 14, Lines 30-32) appears to have a very similar title to the present manuscript and is also a "Part 1". I don't understand the need for a two-part manuscript, and suggest that the authors consider removing these references, removing the Part 1 from the title of the current paper, and incorporating the additional simulation results requested in 1 above into the current paper. If there's a good reason for a two-part paper, then it would be helpful for the authors to provide that rationale in their response and a copy of the other paper so that they can be reviewed together. Often, a stand alone paper followed by a later, independent paper is a better path than a two-part set of manuscripts.

We appreciate the referee's suggestion. In the revised manuscript, we have included the most important model evaluation results from the originally-planned companion paper. We no longer plan to write a second companion paper on this topic due to the unanticipated funding constraints that have already significantly delayed the submission of this revised manuscript.

3) The color scales of the figures are very large and make it difficult to discern small changes in the statistically-significant ocean regions, while emphasizing large changes over the continents. Please mask the continental portions of the maps and change the color scale extents so that the regions of interest are more discernable. Figure 6-topright is a good example of this, where it is hard to tell if LWP varies in the one meaningful shaded region east of Cape Horn by closer to 0% or closer to 60%. Similarly, the accumulation mode number hotspot in east Africa in Fig. 5 dominates that scale at the expense of the North Atlantic and Southern Oceans (where aerosol changes appear C3 to be statistically-significant but close to 0 kg$^{-1}$ ).

Thanks for pointing it out. We have fixed the contours in these figures to improve legibility.

4) Does the model provide CCN information at higher supersaturations than 0.1%, and how do the organic emissions from OCEANFILMS change the shape of the CCN spectrum (i.e., CCN concentration or activated fraction as a function of supersaturation)? How much variability in modeled supersaturation is observed in the model results? Is 0.1% representative of remote regions, or are larger supersaturations found where there are low aerosol concentrations?

The model provides CCN output at S=0.02%, S=0.05%, S=0.1%, S=0.2%, S=0.5%, and S=1.0%. Observed supersaturations in marine boundary-layer clouds vary over approximately the range 0.1% - 1.0%, according to Quinn et al., (2017) and references therein. As shown below, OCEANFILMS (ADD_EXT) increases the zonal mean CCN concentrations at both the lower and upper end of this range.

Change in zonal, seasonal mean CCN at S=0.01%:

[Figure]

Change in zonal, seasonal mean CCN at S=1%:

[Figure]

MOA is expected to have smaller impact on CCN at lower supersaturations and.  The supersaturation 0.1% was chosen for the results shown in the manuscript because it represents the lower end of the range of typical supersaturations in marine boundary layer clouds, and therefore also the

As an example calculation, below is the annual mean CCN spectrum at 50 S, 50W, which is in the region where the effects of OCEANFILMS on CCN are strongest.  CCN is shown with MOA (from the INT_ADD case) and without MOA (from the CNTL case).

[Figure]

5) In Section 2.3, it is stated that the sea spray emissions parameterizations of Martensson et al. (2003) and Monahan (1986) were based on lab experiments that included only inorganic species and no organic species, so it is assumed that these parameterizations do not include any contributions from organic matter. This line of reasoning doesn't make sense to me, as presumably the presence of dissolved organic matter in the seawater would contribute sea spray aerosol emissions from the same bubble bursting processes that produce the inorganic aerosol, but not necessarily in the same amount or size. Monahan's parameterization is limited to particles above 2.5 µm, where perhaps neglecting an organic mass contribution is reasonable, but Martensson et al.'s parameterization extends down to 20 nm, where the likelihood of this is assumption being true appears less convincing. On what basis can it be postulated that the

emission of organic and inorganic species from the same mechanistic process (i.e., bubble bursting) can be treated independently of each other?

Thanks for raising this issue.  On closer examination, we recognize that synthetic seawater concentrates, including the Tropic Marin brand used in the Martensson et al. (2003) study, do contain trace amounts of dissolved organics alongside the inorganic components (Arnold et al., 2009).  Therefore, we have removed the statements regarding this assumption and added a sentence explaining the presence of trace amounts of organics in synthetic seawater.

Arnold, W.R., Cotsifas, J.S., Winter, A.R., Klinck, J.S., Smith, D.S. and Playle, R.C., 2007. Effects of using synthetic sea salts when measuring and modeling copper toxicity in saltwater toxicity tests. *Environmental Toxicology and Chemistry: An International Journal*, *26*(5), pp.935-943.

6) In Section 2.4, some of the key inputs from the biogeochemical model to the OCEANFILMS parameterization are mentioned briefly. While the detailed model discussion is given by Burrows et al. (2014), it would also be helpful to have a few more details given here. For example, which biogeochemistry model was used to compute the monthly mean input fields? What parameters from the E3SM atmosphere model, if any, serve as inputs to the combined OCEANFILMS and sea spray emissions parameterizations (e.g., wind speed)? If OCEANFILMS determines the OMF of the emitted sea spray aerosol, but the sea spray aerosol emissions rate is governed by parameterizations based on no organic emissions, how are these parameterizations blended together in the ADD vs. REMOVE simulations? Is this blending done on a mass or number basis (or even at all - did I misunderstand something here)?

As noted in the response to a previous reviewer (above), we have revised this section to better explain how the ADD and REPLACE sensitivity cases were defined.

The concept of OCEANFILMS is indeed ultimately motivated, in part, by the goal of eventually linking such a model to dynamic representations of ocean biogeochemistry fields, but we felt it would add too much complexity to introduce this dynamic linking before evaluating the model using static input fields, and that a step-by-step approach would be more prudent, in keeping with typical model development best practices.  This approach also mitigates the risks of making the E3SM model representation susceptible to potential errors in the E3SM simulation of ocean biogeochemistry.  Indeed, subsequent work revealed that the earliest release of the E3SM model that included full biogeochemistry capabilities (E3SMv1.1-BGC) exhibited significant biases in simulation of ocean biogeochemistry, which are likely caused primarily by interactions between ocean biological processes and the parameterized physics processes in E3SM's new ocean component model (this is documented in Burrows et al., 2020, JAMES).  With our step-by-step development approach, these known biases in ocean biogeochemistry do not impact the behavior of OCEANFILMS (or onto other aspects of climate or the Earth System in E3SMv1.1-BGC, which has only one-way coupling of climate to ocean biogeochemistry).

We have revised this section to summarize how the input files were developed, and to more clearly explain the dependencies and the relationship to sea spray emissions.  The revised text is as follows (new text bolded):

"The OCEANFILMS parameterization, introduced and described in detail in Burrows et al. (2014), proposes a mechanistic approach for connecting emissions of sea-spray organic aerosol to models of ocean biogeochemistry. **As described previously in Burrows et al. (2014), we used the Parallel Ocean Program (POP; Maltrud et al., 1998) to simulate the ocean's general circulation and its biogeochemical elemental cycling (BEC) routines (Moore et al., 2004) to simulate marine biogeochemistry. Both are components of the Community Earth System Model (CESM; http://www2.cesm.ucar.edu Hurrell et al., 2013). Calculations of the ocean biogeochemistry fields were performed using the CESM 1.0 beta release 11. Because it uses prescribed input files obtained from simulations performed with the earlier POP ocean model, rather than online-simulated biogeochemistry fields, the current implementation of OCEANFILMS in E3SM is not affected by the large biases in prediction of ocean biogeochemistry that have been documented in the first release version of E3SM (Burrows et al., 2020).**

Monthly-mean concentrations of five broad classes of macromolecules in ocean surface waters are derived from **the POP-simulated** distributions of phytoplankton, zooplankton, and semi-labile dissolved organic carbon**, and are provided to E3SM through prescribed input files. The files containing these macromolecular distributions are publicly available as part of the E3SM input data repository (see data availability statement).**

Chemical and physical properties are assigned to each of these macromolecular classes, based on representative proxy molecules for which laboratory measurements are available. Using a Langmuir isotherm-based approach, OCEANFILMS then predicts the surface coverage of ocean bubble films with each of these model macromolecules. This surface film coverage, together with a prescribed bubble film thickness, determines the OMF of the emitted sea spray aerosol, **which is calculated online within E3SM on the basis of the prescribed macromolecule distributions.**

**While OCEANFILMS predicts the OMF solely as a function of the prescribed macromolecule fields, the amount of emitted MOA depends on the combination of the OMF with the emitted sea spray, which is a function of wind speed and sea surface temperature. The application of OMF to the sea spray emissions requires additional assumptions regarding the mixing state and the impact of organic emissions on total emitted particle number and mass, which we explore in four sensitivity cases (described in Section 3.1)."**

7) Section 3.1 currently lacks details on how the ADD versus REMOVE simulations are set up and the meaning of what these simulations represent. For example, am I to understand that if 1 µg of MOA is added to the accumulation mode then 1 µg of NCL is subtracted from the accumulation mode in the REMOVE simulations? Under what circumstances might that make physical sense?

Thanks for pointing this out. In response to this comment and similar comments from the other two reviewers, we have added the following explanation of how the ADD and REPLACE options are implemented, and the physical assumptions under which either the ADD or REPLACE options would make sense.

"Regarding the assumption of replacing versus adding sea salt, we again make two extreme choices. In REPLACE cases, the mass of emitted sea salt is reduced by an amount that is equal to the emitted MOA mass, such that the total emitted aerosol mass remains constant. The number of emitted sea salt particles is also reduced proportionally in each mode. If the underlying sea spray emissions parameterization is assumed to already include the organic content, then the REPLACE option would be the more physically plausible approach to implementing the OMF predicted by OCEANFILMS.

In contrast, if the underlying sea spray emissions parameterization is assumed to include only the inorganic salt components of the emitted spray, then the ADD option would be the more physically plausible approach. In the ADD cases, the mass and number of emitted sea salt are unchanged from the BASE model. Emitted MOA mass is added into the respective aerosol modes, increasing both the total mass and the total number of emissions in that mode."

8) There appear to be large differences in the organic aerosol emissions associated with the ADD vs. the REMOVE simulations (Table 4). Shouldn't at least the sum of the MOA mass emissions rates across all size modes (in Gg/yr) be the same in all 4 simulations? I could see how differences in how these mass emissions are partitioned across mixing state, size, number might drive differences in mean atmospheric burdens (Figure 2 and Table 4), but I don't understand what would change the mass emissions rates themselves. Put another way - if the ocean biogeochemistry model monthly means drive OCEANFILMS, and OCEANFILMS drives the emissions rates, why would these emissions rates vary across the four simulations that incorporate the OCEANFILMS parameterizations?

The total emissions of MOA is indeed very different in the ADD and REPLACE cases. This is a result of the differences in how the organic mass fraction (OMF), which is constant across all four cases, is combined with the sea spray emissions. OMF is given by:

$$OMF = MOA / (MOA + NCL)$$

where MOA and NCL are the mass emissions of marine organic aerosol and sea salt, respectively. Rearranging this equation gives:

$$MOA = NCL / (1/OMF - 1)$$

In the ADD cases, the amount of NCL emitted is held constant, so the emitted MOA is computed by applying this equation to the NCL emissions predicted by the underlying sea spray emissions function.

In contrast, in the REPLACE case, the total sea spray emissions is held constant. Consequently, both the MOA and NCL emissions are proportionally reduced to maintain a constant OMF. This is the reason why the MOA emissions are much smaller in the REPLACE cases, as compared with the ADD cases.

We have added a sentence to the revised manuscript to clarify this: "Note that the source term is significantly higher for the ADD cases than for the REPLACE cases because the total sea spray emissions are modified in the ADD case while holding the organic mass fraction of those emissions constant."

Finally, because these are free-running simulations, minor differences between the sensitivity cases can arise due to differences in simulated meteorology, which changes both the sea spray emissions (due to the dependence of the sea spray parameterization on wind speed) and the loss processes (in particular, due to changes in precipitation rates). This explains the small differences in the Aitken mode emissions across the two ADD and the two REPLACE sensitivity cases.

9) Sections 3.1.1 and 3.1.2 seem out of place here as they provide background from previous observational studies. I think that this discussion would be better suited in the "Introduction" section rather than the "Model simulations and analysis methods" section.

Thanks for this comment; this is a good point. We have moved this material into the "Introduction" section in the revised manuscript.

10) The conclusion drawn on Pg. 15, Lines 4-5 seems very speculative based on the results presented in this manuscript. Despite the good agreement at Mace Head, the comparisons with observational data from Amsterdam Island and Point Reyes don't appear very conclusive. Given the speculativeness of the writing, I suggest that the authors strike this paragraph, and proceed with Sections 4.2 and 4.3 as an example C5 case for closer examination. It might also be worth noting that while the ADD-INT case is presented, the ADD-EXT case would likely yield similar results.

We appreciate the point, and have revised the language here to be more cautious about the conclusions that can be drawn. In the revised text, we now more clearly explain that the ADD_EXT case has been selected as the default for E3SM (which is in part why this case warrants more attention). We explain that this choice was made in part because of the good agreement with observed seasonal cycles (particularly at Mace Head), and in part due to our interpretation of existing observational evidence supporting the ADD and EXT assumptions, but we have softened or removed statements regarding the conclusiveness of the evidence supporting this choice. Finally, following the referee's suggestion, we explicitly point out to readers that the choice of internal versus external mixing is less consequential than the choice of ADD versus REPLACE.

11) I think that the conclusions drawn on Pg. 1, Line 7 and Pg. 20, Line 3 about "best agreement" or "most realistic" are too strong and not supported by the findings of this study. Instead, I suggest that these statements be reworded to note that: the ADD organics simulations show seasonal variation in organic aerosol that is consistent with some past observational studies, while the REMOVE organics simulations lack this pronounced seasonality. The assumption of an internally or externally mixed organic aerosol appears to have a minor, secondary effect on the simulation results (Figure 2, Table 4).

Following up on the reviewer's suggestion, we have clarified and revised these statements as follows along the lines suggested.

12) Since the continents have a greater contribution to the zonal mean in the northern hemisphere than in the southern hemisphere, would recomputing Figure 8 with the continental grid boxes excluded reveal a stronger (larger) period of statistical significance in the Southern Ocean and some period of significance in the North Atlantic and Pacific Oceans?

We agree with the reviewer that recomputing the zonal means with continents excluded might reveal greater regions and periods of statistical significance, particularly in the northern hemisphere, e.g., the North Atlantic. But we're not sure this particular computation is needed for the main analysis and conclusions of this study.

Minor comments:

Pg. 2, Line 11: "of and" missing words

Thanks for catching this; we have fixed the typo.

Pg. 2, Line 27: "ina" typo

Fixed this, thank you!

Pg. 3, Line 26: If this paper is still in preparation, please cite it as such rather than with a publication year.

As explained above, we have decided to add some of the key evaluation results mentioned here into the revised version of this manuscript; we no longer plan to produce a second companion manuscript at this time. We have revised this sentence accordingly, and removed all other references to the originally-planned companion paper.

Pg. 5, Line 29: While "renaming" is the model process of making the jump from one size mode to another, it isn't really a physical process. I suggest removing it from this parenthetical list, but retaining the good, later discussion on Pg. 6, Line 13-14.

Thanks for pointing this out. We have made the suggested change.

Pg. 14, Line 9: Please add a citation for the assumption that OM:OC ~ 1.9.

We thank the reviewer for this comment, which alerted us that we had inadvertently used OM:OC=1.9 for MOA in two figures, while using OM:OC=1.8 elsewhere. For consistency, we have now updated the figures to use OM:OC=1.8 for MOA throughout the entire manuscript.

We have also updated the text to clarify our reasoning for choosing this value. This is the OM:OC ratio of Suwanee River fulvic acid, which is an environmental standard substance that has been used as a proxy for marine organic matter.

It is difficult to identify the most appropriate value for the OM:OC ratio of marine organic matter, since relatively few studies have quantified this ratio for clean marine aerosol, and distinguishing freshly emitted sea spray from aerosol influenced of atmospheric "processing" (i.e., condensation of VOCs and photochemical oxidation processes) is challenging in ambient air. Values ranging from ca. 1.4 to 2.1 have been observed in marine aerosol:

- Hawkins et al. (2010) measured aerosol OM:OC ratios of $2.0 \pm 0.19$, $2.0 \pm 0.11$, $2.1 \pm 0.19$, and $2.0 \pm 0.20$, respectively, using FTIR measurements during four at-sea campaigns.
- Russell et al. (2003) measured OM:OC ratios -- mostly ranging between 1.2 and 1.6, and averaging about 1.4 -- using FT-IR data during the PELTI and ACE-Asia campaigns.
- Saliba et al. (2020) calculated an average OM/OC ratio of 1.5 using the sum of non-refractory from AMS OM and hydroxyl group OM from FTIR, for aerosol sampled in the North Atlantic during the NAAMES field campaign.

We have added two sentences to the manuscript to explain this:

"We have used an organic matter to organic carbon mass ratio (OM:OC) of 1.8 to convert where OC values have been reported; 1.8 is the OM:OC ratio of Suwanee River fulvic acid, which has been used as a proxy for marine organic matter. In the few measurements of OM:OC ratios that have been conducted for marine boundary layer aerosol organic matter, observed values range from at least 1.2 to 2.1 (Russell, 2003; Hawkins et al., 2010; Saliba et al., 2020)."

Hawkins, L.N., Russell, L.M., Covert, D.S., Quinn, P.K. and Bates, T.S., 2010. Carboxylic acids, sulfates, and organosulfates in processed continental organic aerosol over the southeast Pacific Ocean during VOCALS-REx 2008. *Journal of Geophysical Research: Atmospheres*, *115*(D13).

Russell, L.M., 2003. Aerosol organic-mass-to-organic-carbon ratio measurements. *Environmental science & technology*, *37*(13), pp.2982-2987.

Saliba, G., Chen, C.L., Lewis, S., Russell, L.M., Quinn, P.K., Bates, T.S., Bell, T.G., Lawler, M.J., Saltzman, E.S., Sanchez, K.J. and Moore, R., 2020. Seasonal differences and variability of concentrations, chemical composition, and cloud condensation nuclei of marine aerosol over the North Atlantic. *Journal of Geophysical Research: Atmospheres*, *125*(19), p.e2020JD033145.

Pg. 19, Line 28: "sea-spary" typo

Fixed this, thank you!

Pg. 20, Line 29: May wish to update code availability statement if source code was released earlier this year.

The code availability statement has been updated to indicate the archival locations of the E3SMv1.0 code (which includes the OCEANFILMS implementation that was evaluated here within an earlier version of E3SM) and the surface macromolecules input dataset.

Consider reversing Tables 1 and 2, as the abbreviations that are first defined in Table 2 are used in Table 1.

Thanks for this suggestion; we have taken the suggestion and also revised the associated text accordingly.

Table 1: Why does the density of MOA have four significant figures, while the other species have at most 2?

The value of 1601 km m$^3$ is the density of alginic acid; of course, it would not change the answers much to use 1600, but since the values listed here are the exact values that were used in the model implementation, we will keep this in the table. We have added a note in the text to explain that this is the density of alginic acid.

Figure 8: The blue (or black) trace in the top row of panels is very difficult to discern from the green trace. Suggest making it dashed and in front?

These lines are difficult to discern mostly because they are very close to each other. This because the differences in zonal mean SWCF are small relative to the absolute values of the zonal mean SWCF.

In the revised manuscript, we have added additional panels to this figure, showing the zonal mean changes in CCN (S=0.1%), cloud drop number concentration, and cloud liquid water path. Since the relative differences are more prominent for CCN and cloud droplet number concentration, their inclusion should help to make the interpretation of the figure more apparent.

---

## Author Response (AR2)

Dear Susannah and co-authors. Based on the positive feedback from the reviewers, I am happy to accept your revised manuscript. Please, note that the reviewers identified minor typographic errors that need to be fixed:

Thank you for the positive decision.  Please find our point-by-point response to the corrections below.

Page 1, Line 10) I'd suggest using "... reasonably well with observed..."
Done.

Table 2) Numbered superscript within the table doesn't match the lettered superscript at the bottom
Done.

Figure 3) I'd suggest changing the dashed lines representing the observed standard deviation to a lightly shaded area in order to declutter the figure.
Done.

Page 17, Line 8) Remove duplicate period.
Done.

Page 31, Line 24) Add space after period.
Can't find this error at the specified location in the submitted manuscript.